# Random-sequence genetic oligomer pools display an innate potential for ligation and recombination

**Hannes Mutschler[1][†][‡]\*, Alexander I Taylor[1][†], Benjamin T Porebski[1], Alice Lightowlers[1][§], Gillian Houlihan[1], Mikhail Abramov[2], Piet Herdewijn[2], Philipp Holliger[1]\***

[1]MRC Laboratory of Molecular Biology, Cambridge, United Kingdom; [2]REGA Institute, Katholieke Universiteit Leuven, Leuven, Belgium

**Abstract** Recombination, the exchange of information between different genetic polymer strands, is of fundamental importance in biology for genome maintenance and genetic diversification and is mediated by dedicated recombinase enzymes. Here, we describe an innate capacity for non-enzymatic recombination (and ligation) in random-sequence genetic oligomer pools. Specifically, we examine random and semi-random eicosamer ($N_{20}$) pools of RNA, DNA and the unnatural genetic polymers ANA (arabino-), HNA (hexitol-) and AtNA (altritol-nucleic acids). While DNA, ANA and HNA pools proved inert, RNA (and to a lesser extent AtNA) pools displayed diverse modes of spontaneous intermolecular recombination, connecting recombination mechanistically to the vicinal ring cis-diol configuration shared by RNA and AtNA. Thus, the chemical constitution that renders both susceptible to hydrolysis emerges as the fundamental determinant of an innate capacity for recombination, which is shown to promote a concomitant increase in compositional, informational and structural pool complexity and hence evolutionary potential.
DOI: https://doi.org/10.7554/eLife.43022.001

**\*For correspondence:**
Correspondence to: ph1@mrc-lmb.cam.ac.uk; mutschler@biochem.mpg.de

[†]These authors contributed equally to this work

**Present address:** [‡]Max Planck Institute of Biochemistry, Martinsried, Germany; [§]Weatherall Institute of Molecular Medicine, Department of Medicine, University of Oxford, Oxford, United Kingdom

**Competing interests:** The authors declare that no competing interests exist.

## Introduction

The phenotypic richness of a biopolymer pool, that is the density of functional sequences within the sequence space (the totality of all available or possible sequences), is a central, if largely unexplored parameter of its evolutionary potential. This property critically informs many aspects of biological function including the adaptability of biological systems as a function of combinatorial diversity ranging from bacterial populations to the mammalian immune system and to abiotically-generated sequence pools. In all such cases, 'survival' (in a Darwinian sense) depends on the availability of functional entities (such as a neutralizing antibody) within the combinatorial space available to the system (such as the number of circulating B-lymphocytes or the size of a phage display repertoire). The same concept is also of central importance to the proposed spontaneous emergence of function from prebiotic pools of informational polymers, a key conjecture of current scenarios for the origin of life. Indeed, the phenotypic richness of random-sequence pools of the natural nucleic acids RNA and DNA, as well as a variety of xeno nucleic acids (XNAs), has been studied by in vitro selection methods reliant on iterative rounds of enrichment and amplification, yielding ligands (aptamers) and catalysts (*Wachowius et al., 2017*). Extrapolation from such selection experiments and analysis of the mutational landscapes of functional oligomers indicated a low frequency of such functional molecules (1 in $10^{10}$-$10^{13}$) (*Wilson and Szostak, 1999*), suggesting that, globally, the pools are essentially inert, although weak ligands/catalysts might be more frequent (*Ekland et al., 1995*; *Knight et al., 2005*). Together, experimental data and theoretical considerations suggest that function scales with

sequence (i.e. information) and structural complexity (or a related property, termed 'functional information' (*Carothers et al., 2004*)). Thus, sequences with any given function (and among these, those with highest functionality) are more frequently found in longer, informationally and compositionally more complex, and by inference structurally more diverse pools, despite a potential trade-off through increased misfolding (*Ekland et al., 1995*; *Sabeti et al., 1997*; *Carothers et al., 2004*; *Legiewicz et al., 2005*; *Jiménez et al., 2013*). In agreement with this, in silico models predict a sharp drop in the propensity of both folded structures and structural complexity in RNA pools with shorter oligomer lengths (<25 nucleotides, nt) (*Briones et al., 2009*). Consequently, the pools of short, random RNA oligomers of mixed sequence composition (<20 nt) accessible from prebiotic chemistry (*Monnard et al., 2003*) would be predicted to contain few defined three-dimensional structures and even fewer functional sequences. This raises the question how functions that would constitute a living system could ever have emerged spontaneously from such pools.

Starting from the premise that the global potential of oligomer pools for function may have been underestimated by an undue focus on single 'hit' sequences (i.e. fitness peaks), we have sought to examine the innate, global functional potential of the pools themselves, which has never been measured directly. Specifically, we examine pools of short RNA oligomers of semi- and fully-random sequence (with or without activation chemistry) and compare them to equivalent pools of chemically related but distinct genetic polymers, including DNA as well as the xeno nucleic acids (XNAs) ANA (arabino-), HNA (hexitol-) and AtNA (altritol-nucleic acids), with regards to a simple functional test: the ability to undergo intermolecular ligation and/or recombination. While DNA, ANA and HNA pools proved inert, we discover a pervasive tendency of diverse-sequence RNA (and to a lesser extent AtNA) pools to undergo ligation and/or recombination even in the absence of extraneous activation chemistry, and analyse the constitution and emergent properties of recombined pools by deep sequencing, bioinformatic analysis and in silico simulation. Our results suggest that RNA (and to a lesser extent AtNA) hold a privileged position among simple alternatives for genetic information storage and propagation due to an innate capacity for recombination enabled by energetically near-neutral transesterification chemistry (*Eftink and Biltonen, 1983*; *Usher and McHale, 1976*; *Verlander et al., 1973*; *Verlander and Orgel, 1974*). Furthermore, our results demonstrate that such spontaneous, non-enzymatic recombination among random-sequence pool members acts to counterbalance hydrolytic decomposition and in the process, progressively increases pool compositional, informational and structural diversity.

## Results

We chose to specifically examine $10^{15}$-member pools of eicosamer (20 nt long) RNAs. While still within the range of oligomer lengths likely to be accessible via prebiotic chemistry (*Ferris et al., 1996*; *Monnard et al., 2003*), 20-mer sequences are predicted to exhibit at least some tendency for secondary structure formation (*Briones et al., 2009*). Furthermore, a $10^{15}$-member eicosamer pool (1.7 nmol) comprises significant redundancy (909 copies) of each possible 20-mer sequence. Finally, as shown previously, naturally occurring 20-mer sequences can associate into non-covalent multimer complexes reconstituting function such as catalysis, when incubated in eutectic ice phases at −9 °C, as exemplified by the hairpin ribozyme (*Mutschler et al., 2015*; *Vlassov et al., 2004*).

In order to examine the innate potential of such pools for functionality (such asligation), we first examined the reactivity of 2′, 3′-cyclic phosphate (>p)-activated random RNA pools. >p groups form as part of RNA degradation reactions, as well as during proposed prebiotic nucleotide synthesis (*Jarvinen et al., 1991*; *Li and Breaker, 1999*; *Powner et al., 2009*; *Gibard et al., 2018*) and can mediate non-enzymatic ligation in the presence of an organizing template (*Lutay et al., 2006*; *Usher and McHale, 1976*). In order to maximize reactivity, we incubated pools in ice; eutectic ice phase formation has been shown to promote molecular concentration, reduce RNA hydrolysis, and provide a benign, compartmentalized medium, making water ice a potentially beneficial medium for RNA propagation at the origin of life (*Attwater et al., 2010*; *Attwater et al., 2013*; *Mutschler et al., 2015*). We first incubated a >p activated random RNA eicosamer pool ($N_{20}$>p, including 6% 5′-Carboxyfluorescein (FAM) labelled $N_{20}$>p to facilitate detection of ligation products) in ice without any organizing template and sought to detect higher molecular weight (MW) product bands by gel shift in denaturing gel electrophoresis (Urea-PAGE) (*Figure 1A*). We observed significant intermolecular, covalent $N_{20}$>p pool ligation with yields of ~10% (after 2 months, −9 °C) both in

the presence or absence of magnesium (+ / - $Mg^{2+}$) counterions (10 mM $MgCl_2$, *Supplementary file 1*). No ligation was observed in equivalent control samples incubated at $-80$ °C (*Figure 1B*). Deep sequencing of the most prominent higher MW band confirmed that >80% of sequences corresponded to ligation events between two eicosamers ($N_{20} \times N_{20}$, *Figure 1C*). Thus, RNA eicosamer pools activated by >p undergo a disproportionation reaction by intermolecular ligation, yielding a subpopulation of pool sequences extended in length (predominantly 40-mers and some 60-mers (*Figure 1C*)).

Comparison of the 40mer ligation product sequences with the sequence distribution of pre-ligation eicosamers revealed a clear nucleotide signature at the $N_{-1}pN_{+1}$ ligation junction (predominantly CpN, with UpG also overrepresented) (*Figure 1D*, *Figure 1—figure supplement 1A*). While the mechanistic basis for this bias remains to be elucidated, we hypothesize that the stable pairing and the slightly higher reactivity of pyrimidine nucleoside 3′-phosphodiesters (hydroxide ion catalysed cleavage of 3′, 5′-YpN is ca. 2x-3x faster than 3′, 5′-RpN cleavage (*Oivanen et al., 1998*)) aides >p dependent ligation. Regardless of the mechanism, the resulting sequence fingerprint allows an assignment of ligation junctions and reaction trajectories proceeding through an analogous, putative >p like intermediate or transition state in diverse RNA contexts (see below).

Other global features of the ligation junction are notable. In silico RNA folding using RNAfold (*Lorenz et al., 2011*) comparing experimental *versus* computer-generated sequence pools using the same post-ligation nucleotide distribution (shown in *Figure 1D*) indicates a significant enrichment of secondary structure in the ligated pool with ca. 55% of intermolecular ligation junctions located within helical regions (*Figure 1E*). In accordance with previous studies on >p reactivity, this suggests that ligation is accelerated by inter-strand hybridization, which positions reactive groups in proximity and increases their effective concentration, while reducing the entropic cost of intermolecular ligation. In support of this, we found evidence for extensive intramolecular association and non-covalent complex formation in random eicosamer pools as judged by non-denaturing gel electrophoresis (*Figure 1—figure supplement 2*). Thus, random pools contain a significant proportion of organizing templates through intermolecular hybridization among pool oligomers and this provides gapped duplex junctions poised for ligation (*Lutay et al., 2006*; *Usher and McHale, 1976*).

We wondered if these pools contained RNA sequence motifs that would promote ligation by mechanisms other than gapped duplex formation. To aiddiscovery of specific motifs, we repeated the ligation experiments using semi-random RNA sequence pools comprising random eicosamers ($N_{20}$) flanked by (unstructured) 20 nt constant sequence primer binding sites ($C1_{20}$, $C2_{20}$). The use of such 'bait' ($C1_{20}$-$N_{20}$>p, B>p) and 'prey' (5′OH-$N_{20}$-$C2_{20}$, OH-P) oligonucleotides (2.5 µM each) enabled rapid detection and analysis of intermolecular bait $\times$ prey (B $\times$ P) ligation products with high sensitivity by RT-PCR using primers complementary to $C1_{20}$ and $C2_{20}$ as well as much shorter incubation times than needed for direct detection (*Figure 2A,B* and *Figure 2—figure supplement 1A*). Deep sequencing of RT-PCR products revealed ligation of semi-random RNA sequence pools (40 nt) into 80 nt B-P ligation products (5′-$C1_{20}$-$N_{20}$-$N_{20}$-$C2_{20}$-3′), with very similar features as in the fully-random pool, including the above described CpN ligation junction 'fingerprint' and the increased base-pairing bias at the ligation site (*Figure 1—figure supplement 1B*, *Figure 2—figure supplement 1B*). Indeed, when we randomly sampled clones from the ligated fraction and analysed the ability of the constituent B, P segments of the most active clone for ligation, activity depended on formation of a gapped duplex junction with distal sequences having little impact on ligation (*Figure 2—figure supplement 2*).

Applying a more stratified analysis, we searched for other RNA motifs than a gapped duplex that may mediate RNA ligation by comparing ligated B-P-sequences from two different semi-random RNA pools (differing in their $C_{20}$ sequences, *Supplementary file 2*). The use of different $C_{20}$ anchor sequences ($C3_{20}$, $C4_{20}$) allowed us to dissociate motif structure from sequence-dependent effects such as the imprint of pairing with the conserved $C_{20}$ termini. We clustered ligation junction structures according to (predicted) minimal free energy using 408 different secondary structure sub-motifs based on 25 main-motifs (Motif A - Motif Y) including internal loops and hairpins (*Figure 2—figure supplement 3*). While ligation frequencies of most of these motifs broadly correlated with predicted base pairing frequency at the ligation junction (consistent with formation of a helical junction and proximity ligation, see above), we identified high ligation frequencies at specific positions in the unpaired regions of some predicted internal loop motifs (motif H and J, *Figure 2C,D*). The consensus sequence of these submotifs, H4 and J4, could subsequently be extracted using the motif

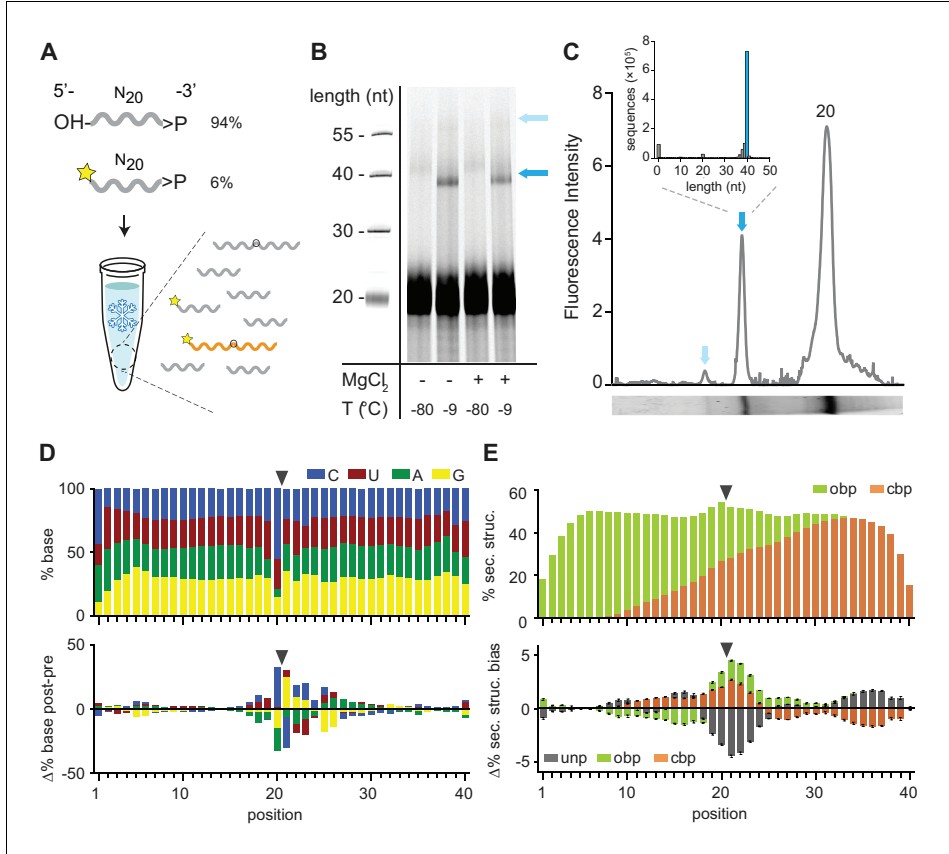

**Figure 1.** Reactivity of 2', 3' cyclic phosphate (>p) activated $N_{20}$ RNA pools. (**A**) $N_{20}$>p pools are incubated in eutectic ice phases, spiked with 6% 5'FAM (Carboxyfluorescein)-labelled $N_{20}$>p to facilitate detection of ligation products (orange) by Urea-PAGE. (**B**) Scan of a denaturing gel for FAM-labelled products after 57 days of incubation either in eutectic ice ($-9\ °C$) or at $-80\ °C$, and +/- $MgCl_2$. Note that ligation of FAM-$N_{20}$>p is inhibited due to the blocked 5'-OH. (**C**) Densitogram of a SYBR-gold stained Urea-PAGE gel trace (-$MgCl_2$, $-9\ °C$) (bottom panel) showing total ligation (both FAM-labelled and unlabelled ligation products (indicated by arrows)). The size distribution and size of the main ligation product (40mer) from deep sequencing indicates direct ligation of two eicosamers (inset). (**D**) Upper panel: Average nucleotide distribution profile of the 40mer ligation products. Lower panel: Changes in nucleotide composition compared to the unligated input $N_{20}$>p pool. Black arrows indicate the ligation site. (**E**) Upper panel: Average base-pairing frequencies of sequenced 40mers (as predicted by RNAfold (*Lorenz et al., 2011*): 'Opening' base pairs (obp, green; defined as hybridization to a downstream nucleotide) and 'closing' base pairs (cpb, hybridization to an upstream nucleotide, orange). Lower panel: Average difference of predicted base pairing between experimental pool and synthetic sequence pools (N = 3; 300,000 sequences each) generated in silico using experimental nucleotide frequencies shown in panel D. Grey indicates predicted unpaired bases (unp), black standard deviations.

DOI: https://doi.org/10.7554/eLife.43022.002

The following source data and figure supplements are available for figure 1:

**Source data 1.** Source data for panels C, D, and E.
DOI: https://doi.org/10.7554/eLife.43022.006

**Figure supplement 1.** $N_{-1}pN_{+1}$ dinucleotide frequencies.
DOI: https://doi.org/10.7554/eLife.43022.003

**Figure supplement 1—source data 1.** NpN frequencies of different ligation products and relative change to pre-ligation material.
DOI: https://doi.org/10.7554/eLife.43022.004

**Figure supplement 2.** Evidence for intramolecular association and non-covalent complex formation in pools of random RNA oligonucleotides as judged by gel electrophoresis.
DOI: https://doi.org/10.7554/eLife.43022.005

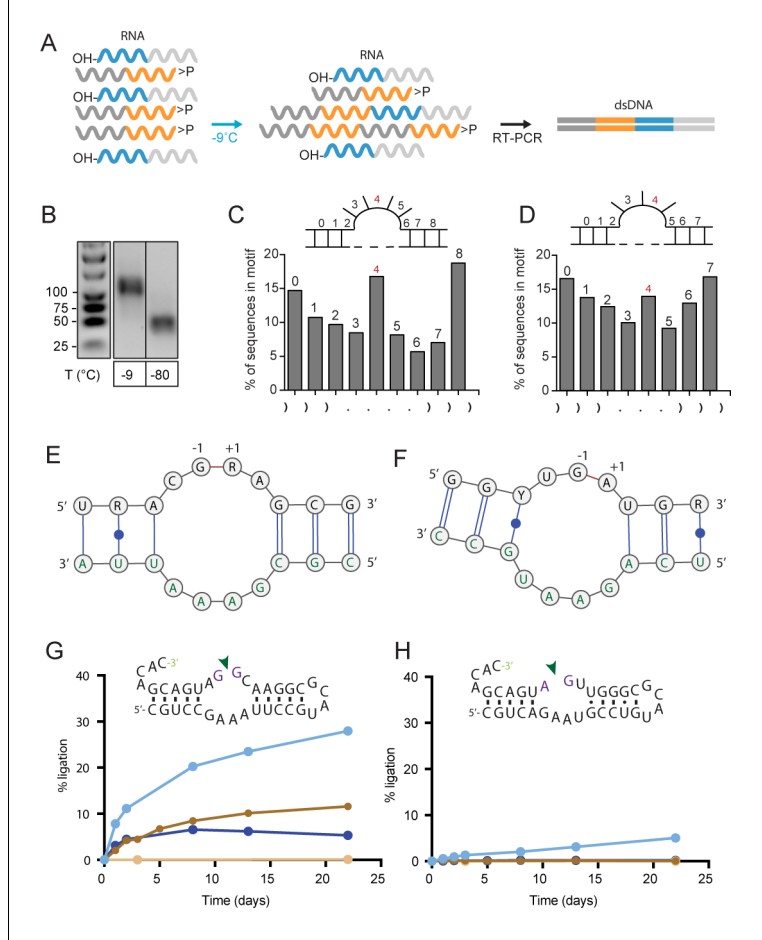

**Figure 2.** Reactivity of semi-random RNA pools with different termini. (**A**) Assembly scheme: The random $N_{20}$ regions of both semi-random RNAs are illustrated in orange (bait) and cyan (prey). Only direct ligation products comprising the constant primer binding site segments 5'-$C1_{20}$ (dark grey) and 3'-$C2_{20}$ (light grey) are detected by RT-PCR. (**B**) RT-PCR products for bait>p × 5'OH-prey reactions after incubation at −9 °C or −80 °C for 51 days. Shorter PCR products recovered at −80 °C result from unspecific annealing of the RT-primer in the 3'-random region of the bait fragment (*Figure 2—figure supplement 1A*). (**C, D**) Ligation frequencies for motif J (**C**) respectively motif H (**D**) in bait>p × 5'OH-prey products. Motif J is defined by four unpaired bases flanked by 3 nt helical regions. The plot shows the normalized amount of ligation sites located between two adjacent nucleotides within motif J. Although the ligation frequency generally drops in regions predicted to be unpaired, in submotifs J4 and H4 it is enriched (shown in red). (**E**) Consensus motif for J4 extracted from the sequence peak in (**C**). The constant regions form the strand opposite of the ligation site (green letters). (**F**) Consensus motif for H4. As for J4, the ligation site comprises parts of the constant region (green). (**G**) A minimal RNA construct with the J4 core-motif catalyses bimolecular ligation under frozen conditions in ice (-$MgCl_2$, cyan; +10 mM $MgCl_2$, dark blue) as well as under ambient temperature (17 °C) in presence of $MgCl_2$ (brown). No ligation under ambient conditions is observed in absence of $MgCl_2$ (light brown). (**H**) H4 catalyses slow 3'−5' ligation reactions exclusively under frozen conditions and in absence of $Mg^{2+}$ (blue).

DOI: https://doi.org/10.7554/eLife.43022.007

The following source data and figure supplements are available for figure 2:

**Source data 1.** Source data for panel C, D, G, and H.
DOI: https://doi.org/10.7554/eLife.43022.015

**Figure supplement 1.** Detection of ligation of >p activated semi-random sequence pool RNA by RT-PCR.
DOI: https://doi.org/10.7554/eLife.43022.008

**Figure supplement 1—source data 1.** Source data for panel B, C, and D as well as pre-ligation frequencies of all semi-random RNA pools used for recombination/ligation.
DOI: https://doi.org/10.7554/eLife.43022.009

*Figure 2 continued on next page*

*Figure 2 continued*

**Figure supplement 2.** Characterization of four randomly picked clones from semi-random sequence RNA pool experiments.
DOI: https://doi.org/10.7554/eLife.43022.010

**Figure supplement 3.** Secondary structure motifs used to classify RNA ligation products according to predicted secondary structures proximal to the ligation junction.
DOI: https://doi.org/10.7554/eLife.43022.011

**Figure supplement 4.** Characterization of ligation activity of minimized versions of H4, J4 (J4-min, H4-min) or a derived gapped duplex (H4-splint-min) RNA (**Supplementary file 2**) under different conditions.
DOI: https://doi.org/10.7554/eLife.43022.012

**Figure supplement 5.** Analysis of regioselectivity of ligation by H4 and J4 RNA motifs.
DOI: https://doi.org/10.7554/eLife.43022.013

**Figure supplement 6.** Apparent ligation rates of the minimized versions J4-min, H4-min and representative gapped duplexes (H4-splint-min, CCU-splint-min) under different conditions.
DOI: https://doi.org/10.7554/eLife.43022.014

discovery tool DREME (**Bailey, 2011**). In both motifs, internal loop segments are formed opposite the putative ligation junctions with involvement of sequences from the $C_{20}$ constant regions, which seemed to enhance ligation (and therefore enrichment) of compatible junctions from a subpopulation of random segments (**Figure 2E,F**).

Strikingly, motif J4 and H4 were sufficient to promote regioselective ligation in minimal hairpin constructs (**Figure 2G,H**, **Figure 2—figure supplements 4** and **5**): A minimized version of the J4 motif promoted reversible ligation via 2′−5′ regiochemistry both in ice (in the absence of $Mg^{2+}$) and at ambient conditions (strictly $Mg^{2+}$-dependent) with an up to 10-fold enhanced apparent ligation rate ($k_{obs}$) compared to a simple gapped duplex reaction under the same conditions (**Figure 2—figure supplement 6**). A search for the J4 consensus motif in the JAR3D structure database (**Zirbel et al., 2015**) revealed that J4-like motifs are present in in naturally occurring RNAs such as the domain II of the 23S rRNA of *Deinococcus radiodurans* (**Harms et al., 2001**) and the *Escherichia coli* adenosylcobalamin riboswitch (**Johnson et al., 2012**), for which structural information is available. In both structures, the J4-like motifs form a stable pseudohelical structure comprising a purine-rich 4 × 4 internal loop with a triple-sheared GA motif that immediately suggests a likely catalytic involvement of a cross-strand adenine in the reversible 2′−5′ ligation reaction (**Figure 3**). By contrast, no structural information is currently available for the H4 motif, which mediates ligation via canonical 3′−5′ regiochemistry (exclusively under eutectic ice conditions (−9 °C) in the absence of $Mg^{2+}$) albeit with a 2- to 3-fold slower $k_{obs}$ than an equivalent gapped duplex-mediated reaction (**Figure 2—figure supplement 4**, **Figure 2—figure supplement 6**). Thus, random pools comprise a wide variety of RNA motifs capable of promoting >p ligation via either 2′−5′ or 3′−5′ regiochemistry.

Next, we sought to dissect mechanistic modes of intermolecular RNA ligation using binary combinations of semi-random RNA bait x prey (B × P) sequence pools (2.5 µM each) each bearing different 2′, 3′- or 5′-substituents (**Figure 4A**). As described above, we observed efficient and rapid ligation of B>p × 5′OH-P pools (and no ligation in negative controls incubated at −80 °C), as judged by RT-PCR (**Figure 4B**). Furthermore, we also observed intermolecular assembly in 2′- (or 3′-) phosphorylated B-2′p (or B-3′p) × 5′OH-P reactions. However, these predominantly yielded shorter products, varying between 40–80 nt in length (**Figure 4B,C**), while still displaying the characteristic CpN ligation 'fingerprint' suggestive of reaction via a >p intermediate (**Figure 4D**, **Figure 1—figure supplement 1E**). Thus, the CpN sequence signature allowed mapping of presumed ligation junctions, which revealed variable-length assembly products derived from progressive truncation of bait segments, while preserving prey segment length (**Figure 4D**, **Video 1**), suggesting a mechanism involving intermolecular recombination (rather than ligation). Recombination likely proceeds via a transesterification mechanism involving a nucleophilic attack of the prey 5′-OH on bait segments either directly (leading to internal bait segment cleavage) or on the >p of bait segments truncated by hydrolysis (**Figure 4—figure supplement 1**). Consistent with this mechanistic hypothesis, blocking the prey pool 5′-OH by phosphorylation (5′p-P, **Figure 4A,B**) abolished recombination (**Figure 4B,C**). Furthermore, qRT-PCR suggests that recombination proceeds with a >10 fold slower

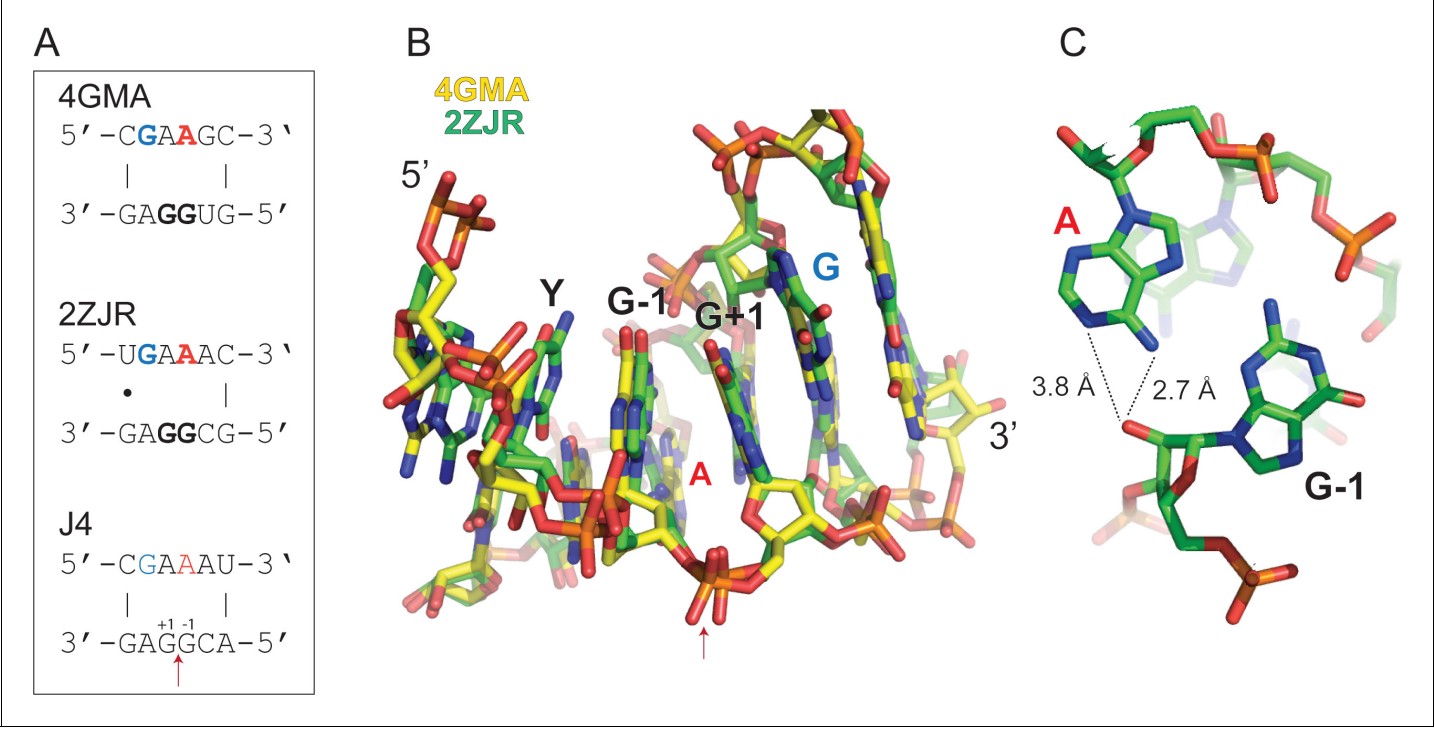

**Figure 3.** Structure of J4-like motifs that occur in natural RNAs. (**A**) J4 likely forms a purine-rich 4 × 4 internal loop classified as a triple-sheared GA motif (RNA 3D Motif Atlas id IL_93568.6) by JAR3D (*Zirbel et al., 2015*), which also occurs naturally in the 23S rRNA of *Deinococcus radiodurans* (*Harms et al., 2008*) (PDB ID: 2ZJR) and the *Escherichia coli* adenosylcobalamin riboswitch (*Johnson et al., 2012*) (PDB ID: 4GMA). (**B**) Consecutive GA base pairs in such internal loops have a strong stabilizing effect on the pseudohelical structure (*Chen and Turner, 2006*). Shown are the consensus motif (top) and the PDB overlay of the 4GMA (yellow) and 2ZJR (green) substructures. Note, the canonical 3′–5′ linkage between the $G_{-1}$ and $G_{+1}$ equivalents in 4GMA and 2ZJR is a 2′–5′ linkage in ligated J4. (**C**) In 2ZJR, the central A (red) of the GAAG coding strand forms a sheared AG base pair with the first G of the opposite CGGA motif ($G_{-1}$ in J4) indicating that a similar conformation could exist in J4. The adenine moiety of the central A (red) is in close proximity to the 2′OH of the $G_{-1}$ equivalent suggesting a potential catalytic activation of the 2′OH nucleophile in the reversible transesterification of the 2′–5′ phosphodiester bond between $G_{+1}$ and $G_{-1}$ in J4 through H-bonding.
DOI: https://doi.org/10.7554/eLife.43022.016

rate than direct >p dependent ligation (*Figure 4—figure supplement 2*), consistent with a putative multi-step reaction trajectory involving a slow, hydrolytic generation of truncated, activated bait (B>p) intermediates for ligation, as well as potentially direct in-line attack in a subset of reactions (*Figure 4—figure supplement 1*).

In order to explore the consequence of alternative activating groups, prey pool activation with a 5′-triphosphate (5′ppp-P) was also examined (*Figure 4—figure supplement 3*). These yielded an even broader length distribution of products, comprising full-length 80 nt B-P ligation products (likely originating by direct nucleophilic attack of the bait pool terminal bait-3′OH (or −2′OH) attack on the 5′ppp of the prey pool), as well as a wide range of shorter products, derived from both B and P truncationlikely as a result of non-canonical, multi-step reaction trajectories. Indeed, we note that our method of RT-PCR amplification omits the detection of any potential intermolecular recombination or ligation product that does not display a B-P sequence arrangement. Therefore, bait or prey-pool self-ligation (or recombination) (yielding B-B, P-P type products) or reverse ligation (yielding P-B) or indeed non-canonical intramolecular reactions yielding circular, lariat or branched products (unless resolved into a B-P arrangement) are not detected (*Figure 4—figure supplement 4*). Thus, although rapid and convenient, our PCR assay likely underestimates the true level of intra- and inter-pool reactivity (and the true molecular diversity of reaction products).

Nevertheless, most significantly (and consistent with the reactivities observed above), we observed comparable intermolecular reactivity in unmodified semi-random RNA sequence bait and prey pools ($C1_{20}$-$N_{20}$-3′OH × 5′OH-$N_{20}$-$C2_{20}$) - that is pure RNA pools devoid of phosphorylation or any other activation chemistry. Remarkably, these showed intermolecular assembly, yielding a similar

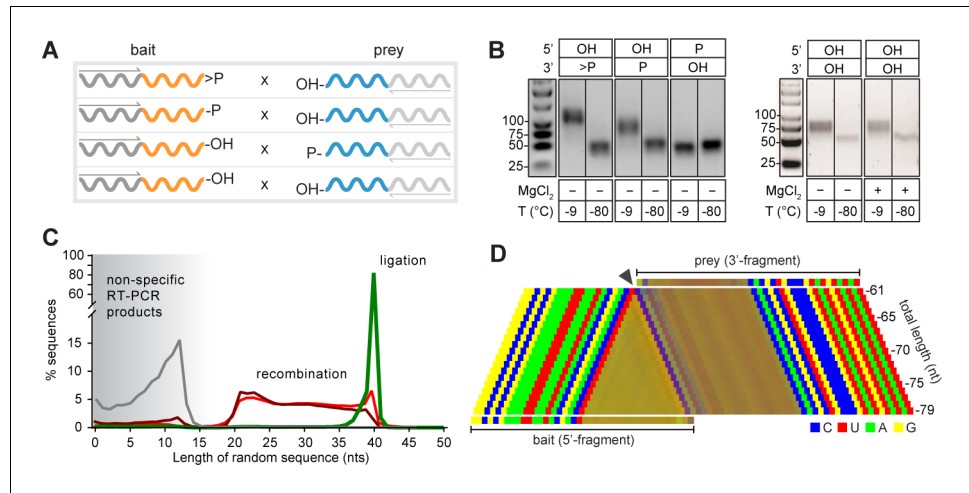

**Figure 4.** Reactivity of semi-random sequence RNA pools with different 2', 3'- or 5'-substituents **(A)** Binary combinations of semi-random sequence bait and prey RNA pools and their reactivity as visualised by RT-PCR **(B)** after incubation at −9 °C or −80 °C (51 days). As in *Figure 2B*, smaller PCR products recovered at −80 °C **(C)** result from unspecific annealing of the RT-primer in the 3'-random region of the bait fragment. **(C)** Product size distributions from deep sequencing of bait>p × 5'OH-prey (green), bait-2'/3'p × 5'OH-prey (dark red), bait-OH × 5'p-prey (grey) and bait-OH × 5'OH-prey (light red) co-incubation reactions. **(D)** Nucleotide signature for products created by recombination of bait-2'/3'p × 5'OH-prey. The frequency of the four nucleotides is represented by a linear combination of RGB values (C-blue, U-red, A-green, G-yellow) resulting in mixed colours in random segments (Material and methods). The characteristic CpN signature at the presumed recombination site (arrow) indicates that the 5'-end of the full-length prey ligates to progressively truncated fragments of >p-activated bait suggesting recombination between bait and prey RNAs in both 2', 3' phosphorylated (bait-2'/3'p × 5'OH-prey) and unmodified RNA (bait-OH × 5'OH-bait) reactions.
DOI: https://doi.org/10.7554/eLife.43022.017

The following source data and figure supplements are available for figure 4:

**Source data 1.** Source data for panel C.
DOI: https://doi.org/10.7554/eLife.43022.024

**Figure supplement 1.** Proposed mechanisms of ligation/recombination in semi-random sequence RNA pools.
DOI: https://doi.org/10.7554/eLife.43022.018

**Figure supplement 2.** Spontaneous recombination in semi-random pools is much slower than direct >p activated ligation.
DOI: https://doi.org/10.7554/eLife.43022.019

**Figure supplement 2—source data 1.** qRT-PCR data of >p ligation vs recombination.
DOI: https://doi.org/10.7554/eLife.43022.020

**Figure supplement 3.** Reactivity of 5'ppp activated semi-random sequence RNA pools under eutectic conditions.
DOI: https://doi.org/10.7554/eLife.43022.021

**Figure supplement 3—source data 2.** Source data for panel B.
DOI: https://doi.org/10.7554/eLife.43022.022

**Figure supplement 4.** Hypothetical ligation/recombination products that can form in semi-random RNA pools but remain undetectable by RT-PCR.
DOI: https://doi.org/10.7554/eLife.43022.023

distribution of truncated products as observed for bait-2'/3'p×5'OH prey reactions, suggesting intermolecular recombination via transesterification chemistry as outlined above (*Figure 4B,C*).

In order to better understand the reactivity of unmodified RNA pools without the obscuring imprint of conserved $C_{20}$ sequences (and PCR amplification bias), we next sought to directly observe reactivity of random RNA eicosamer pools devoid of any activation chemistry. In these (bait × prey: 5'FAM-$N_{20}$-3'OH × 5'OH-$N_{20}A_{10}$-3'OH), the prey pool included an invariant $A_{10}$ tail sequence to aid recovery and analysis (by poly-dT primed reverse transcription), and to allow detection of prey self-recombination products (P-P). After an extended incubation (5 months, −9 °C) we were able to directly visualize formation of two distinct recombination products by gel electrophoresis without

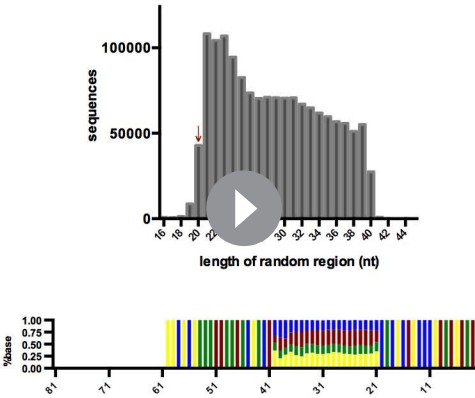

**Video 1.** Animated scan through all sequenced products of different length of a $C1_{20}-N_{20}-2'/3'p$ (bait) × $5'OH-N_{20}-C2_{20}$ (prey) recombination reaction. The nucleotide signature of the different-length products indicate that they were predominantly formed through ligation of full-length prey to truncated and therefore >p activated bait segments.
DOI: https://doi.org/10.7554/eLife.43022.025

any amplification (*Figure 5A*). The two main product bands (Rec1 and Rec2, together ca. 2.3% yield) were analysed using either poly-dT-primed (for Rec1) or poly-U tailing followed by poly-dA primed SMARTer RT-PCR (Rec1 and Rec2) and deep sequencing (*Figure 5—figure supplement 1*). For Rec1, we found recombination fragment distributions from ~40 to ~50 nt that include the characteristic fingerprint of a >p ligation junction (*Figure 5B,C*). In contrast, the longer (on average) Rec2 products (~45 to ~55 nt) showed an ApN ligation junction as they were formed (albeit less efficiently (ca. 0.2% yield)) from recombination of two prey segments within the $A_{10}$ tail (*Figure 5B,C*). Thus, as with the unmodified semi-random sequence RNA pools (*Figure 4B,C*), intermolecular ligation products in random, non-activated RNA pools are likely formed by a transesterification process initiated either by direct recombination or by hydrolytic cleavage (of either B or P), followed by putative ligation of truncated RNA>p sub-fragments. These mechanisms give rise to recombination product species such as bait-prey (B-P, forming Rec1) or prey-prey (P-P present in both Rec1 and Rec2). Bait-bait recombination (B-B) products are not observed, as the bait 5'OH is blocked by FAM modification to aid analysis.

Comparison of Rec1 sequence pools with three random pools (3 × 300,000 sequences) of equal nucleotide distribution generated in silico again indicated that recombination sites were preferentially localized in base-paired (gapped duplex motif) regions (*Figure 5D*). Furthermore, selected pools not only exhibited an overall increase in local duplex formation, but also generally more stable (predicted) minimal free energy structures (*Figure 5E*). Thus, pure random-sequence RNA pools, devoid of any activation chemistry or conserved sequence regions (apart from an $A_{10}$ tag) display an innate capacity for intermolecular recombination, yielding subsets with altered oligomer length distribution (including longer RNA oligomers), enhanced sequence diversity and structural stability.

Finally, we sought to dissect critical chemical parameters required for an innate recombination potential. To this end, we expanded our investigation beyond RNA to compare and contrast the potential of pools of a range of RNA congeners (representing systematic variations of the canonical ribofuranose ring structure), with respect to intermolecular ligation or recombination. We first examined unmodified, semi-random sequence pools of DNA or prebiotically plausible ANA (arabino nucleic acid) (*Noronha et al., 2000*; *Roberts et al., 2018*). In these chemistries, the canonical ribofuranose ring of RNA is replaced with either a 2'-deoxyribose (DNA) lacking the 2' hydroxyl of RNA, or an arabinofuranose ring (ANA), in which the analogous 2'-hydroxyl group is in the axial (*trans*) position. Like RNA, both DNA and ANA are capable of forming functional catalytic motifs (*Silverman, 2016*; *Taylor et al., 2015*), including small self-cleaving DNA catalysts (*Gu et al., 2013*). Therefore, both DNA and ANA pools contain sequences with potential function, and in the case of DNA should be able to perform autocatalytic DNA strand cleavage, as a first step of a recombination reaction. However, while we readily detected recombination in the semi-random-sequence RNA pools (as above), neither DNA nor ANA pools showed detectable recombination even after prolonged incubation (up to 6 months, −9 °C; *Figure 6A*) and despite the 10–100-fold higher sensitivity of PCR (DNA) compared to RT-PCR (RNA and ANA) detection (*Ohuchi et al., 1998*; *Okello et al., 2010*). Even inclusion of transition metal ions ($Zn^{2+}$), which had been shown to strongly promote DNA self-cleavage (*Gu et al., 2013*), did not result in recombination in DNA (or ANA) pools, while apparently inhibiting it in RNA pools (*Figure 6—figure supplement 1*).

Next, we explored the reactivity of pools of two additional RNA congeners, in which the five-membered ribofuranose ring is replaced by six-membered ring analogues: D-altritol nucleic acid

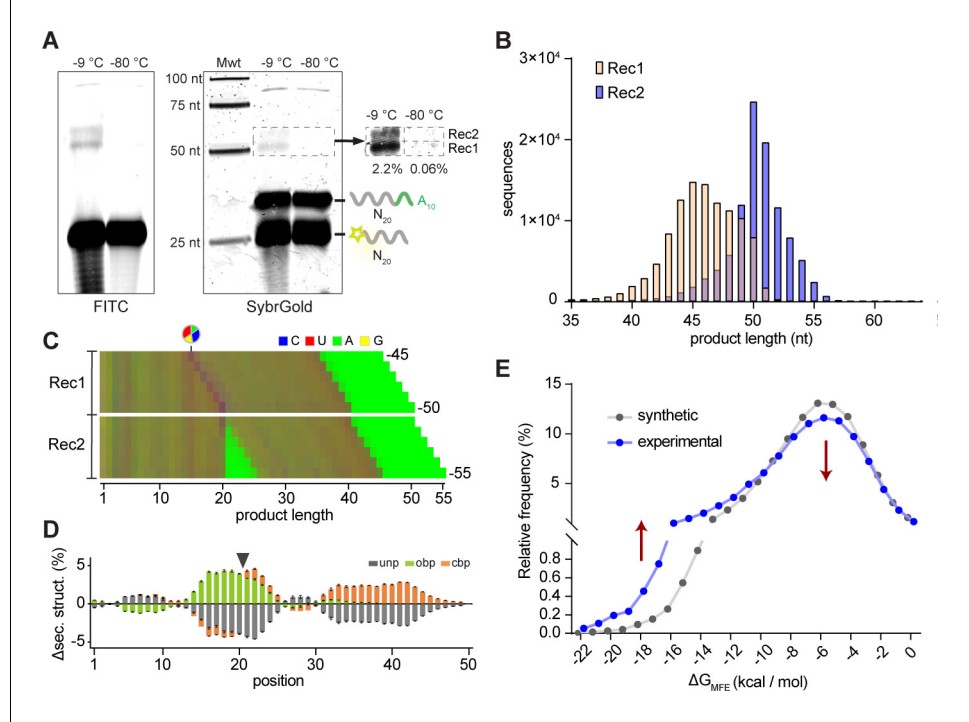

**Figure 5.** Reactivity of unmodified random RNA pools. (**A**). Denaturing gel electrophoresis of unmodified $N_{20}A_{10}$ and FAM-$N_{20}$ RNA pools after incubation ($-9\,°C$ or $-80\,°C$, 5 months) imaged both for FITC fluorescence (showing FAM-$N_{20}$ and its products) (left panel) and after SYBRGold staining (showing all RNAs). Two main product bands (Rec1, Rec2) are apparent. Note the RNA degradation products arising upon prolonged -9 °C but not -80 °C incubation. (**B**) Size distributions of the sequenced Rec1 (yellow) and Rec2 (blue) bands. (**C**) Color-coded nucleotide frequencies of Rec1 and Rec2 sequences recovered (according to the protocol shown in *Figure 5—figure supplement 1A,B*). (**D**) Average base-pairing differences between the experimental 50mer product pool from Rec1 and randomly generated in silico sequence pools (N = 3; 300,000 sequences each) using the experimental nucleotide frequencies from (**C**) with secondary structure calculated using RNAfold. Grey indicates predicted unpaired (unp) nucleotides, green nucleotides involved in opening base pairs (obp) and orange closing base pairs (cbp). Standard deviations are shown in black. (**E**) Predicted energies of the minimal free energy (MFE) structures from the three experimental and in silico pools (as in (**D**)) suggest that recombination product pools show enhanced folding stabilities. Note that the error bars for the energies calculated from the three synthetic datasets are too small to be displayed.

DOI: https://doi.org/10.7554/eLife.43022.026

The following source data and figure supplements are available for figure 5:

**Source data 1.** Source data for panel B, D, and E.

DOI: https://doi.org/10.7554/eLife.43022.029

**Figure supplement 1.** Recovery of products from unmodified RNA $N_{20}$ / $N_{20}A_{10}$ recombination reactions (*Figure 5*).

DOI: https://doi.org/10.7554/eLife.43022.027

**Figure supplement 1—source data 1.** Source data for panel C.

DOI: https://doi.org/10.7554/eLife.43022.028

(AtNA), which shares an analogous vicinal 2′, 3′-cis-diol configuration with RNA (*Figure 6B*), and its 2′-deoxy-analogue, hexitol nucleic acid (HNA). While HNA random-sequence pools had been shown to contain catalytic motifs (*Taylor et al., 2015*), AtNA random-pools had not previously been explored.

To test whether the AtNA vicinal hydroxyl can participate in transesterification reactions despite its different ring geometry, we first exposed defined-sequence HNA and AtNA (as well as equivalent RNA) oligonucleotides to alkaline pH (NaOH). Indeed, while HNA proved inert, AtNA (like RNA) proved to be sensitive to alkaline pH-induced hydrolysis, albeit with an apparent 6-fold slower strand

scission rate than RNA (*Figure 6—figure supplement 2*). This indicates that (akin to RNA) the equivalent vicinal 2′-OH in AtNA can participate in intramolecular transesterification reactions presumably via an analogous cyclic phosphate (AtNA>p) intermediate (*Figure 6—figure supplement 3*). At the same time, the observed lower hydrolytic reactivity suggests that formation of AtNA>p cyclic phosphate may be energetically less favourable than in RNA. Next, we examined whether HNA and AtNA pools (devoid of extraneous activation chemistry) could participate in intermolecular recombination reactions. To this end, we compared the reactivity of semi-random sequence pools of HNA and AtNA to equivalent RNA pools, but observed unambiguous recombination only in the RNA pools (as seen previously) (*Figure 6—figure supplement 4*). To expedite potentially weak tendencies for recombination, we exposed semi-random sequence pools of HNA and AtNA (as well as control RNA) pools briefly to alkaline pH (50 mM NaOH at 65 °C (RNA (3 min), HNA/AtNA (27 min), neutralized with 1 M Tris•HCl pH 7.4) in order to generate an increased amount of >p products, before a prolonged incubation (−9 °C ice, 5 months). Using a novel engineered reverse transcriptase (RT TK$^2$, see Materials and methods), which, unlike commercial enzymes, is able to reverse transcribe both RNA as well as both HNA and AtNA oligomers, we analysed ligation products by RT-PCR. While HNA pools again proved inert, we now observed clear intermolecular recombination in the AtNA pools, although to a notably lesser extent than equivalent RNA pools.

Deep sequencing of the recombination products from both RNA and AtNA pools revealed a similar size distribution indicative of recombination but, in the case of RNA, biased towards smaller products, presumably due to increased hydrolytic degradation induced by exposure to alkaline pH (*Figure 6C*). Furthermore, a clear ApN nucleotide signature was observed at the AtNA recombination junction, distinct from the CpN signature of RNA, presumably reflecting the different chemical requirements for AtNA recombination (*Figure 6D*). The finding that only RNA (and to a lesser extent AtNA) pools are capable of intermolecular recombination, while DNA, ANA and HNA pools were inert, strongly suggests that a vicinal cis-diol configuration forms a critical chemical determinant of an innate intermolecular recombination potential with the propensity for recombination modulated by ring geometry.

## Discussion

Recombination is a fundamental process in biology both central to genetic diversification and genome repair (*Pesce et al., 2016*; *de Visser and Elena, 2007*), as well as a protective mechanism against the progressive corruption of genetic information by drift (i.e. Muller's ratchet (*Muller, 1964*)). RNA recombination is widespread among RNA viruses (*Simon-Loriere and Holmes, 2011*); while generally attributed to template switching, recombination by purely chemical processes has been suggested to occur in polio (*Gmyl et al., 1999*) and rubella viruses (*Adams et al., 2003*), as well as Qbeta bacteriophage (*Chetverin et al., 1997*).

Here, we show that a capacity for spontaneous ligation and recombination is an innate property of random RNA (and to a lesser extent AtNA) oligomer pools and requires neither recombinase enzymes nor extraneous activating chemistry. As we demonstrate, such non-enzymatic recombination and ligation involves, to some degree, specific sequence motifs (e.g. in RNA pools) that promote ligation with either 2′−5′ or 3′−5′ regioselectivity (*Figure 2*, *Figure 2—figure supplement 5*). However, the bulk of observed reactivity derives from the spontaneous organization of random sequence oligomers into non-covalent complexes through intermolecular hybridization, which can be visualized by native gel electrophoresis (*Figure 1—figure supplement 2*).There is staggering potential combinatorial diversity of such intra-pool interactions. Within the $N_{20}$ pools examined herein (with a potential sequence diversity of ca. $10^{12}$), such combinatorial diversity amounts to $>10^{24}$ potential binary and an even vaster number of potential tertiary or higher order interactions. In the case of random RNA as well as semi-random RNA and AtNA oligomer pools, a subset of these non-covalent intermolecular assemblies do not remain inert (and non-covalent), but display a pervasive tendency to undergo spontaneous disproportionation reactions resulting in intermolecular recombination and ligation.

The discrepancy between the extent of ligation / recombination we observe (2–10 % of pool sequences (*Figure 1*, *Figure 4*) to the predicted frequency of functional sequences of, for example, aptamers / ribozymesfrom repertoire selection / SELEX experiments (1 in $10^{10}$-$10^{13}$ (*Wilson and Szostak, 1999*)) should be viewed within the same context. Indeed, standard SELEX experiments

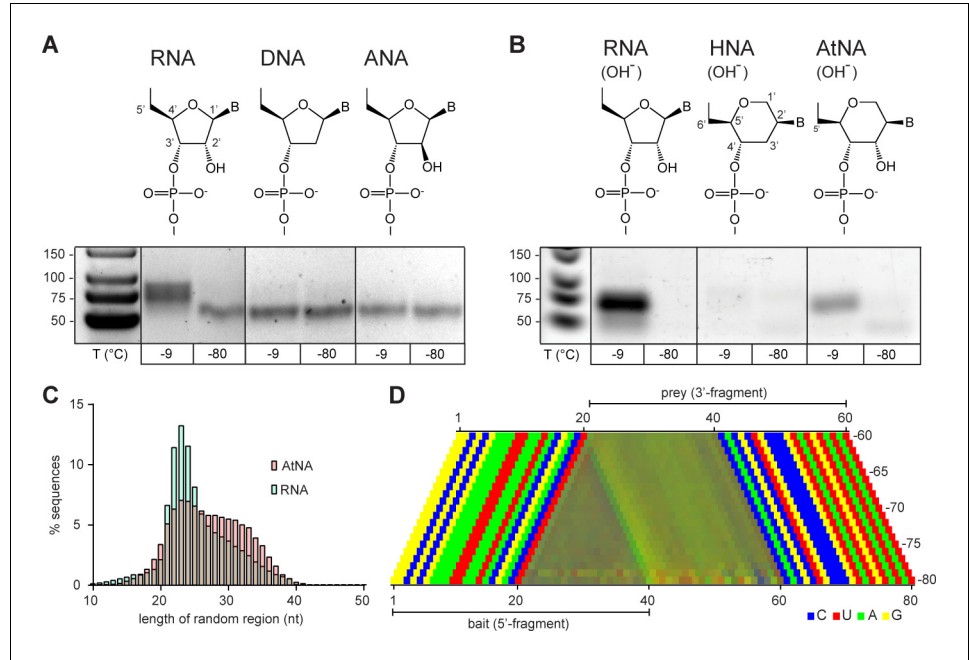

**Figure 6.** Reactivity of unmodified semi-random sequence pools of RNA and related genetic polymers. Binary combinations of semi-random sequence bait and prey pools of RNA, DNA, ANA, HNA and AtNA and their reactivity are visualised by RT PCR as in *Figure 4*. (**A**) In contrast to semi-random sequence pools of RNA (left panel), DNA and ANA pools show no detectable recombination (−9 °C, 5 months). (**B**) Unmodified semi-random sequence RNA and AtNA pools show recombination after a brief exposure to alkaline pH, while HNA does not. Note that the typical low-molecular weight band seen in −80 °C negative control reactions is reduced or absent potentially due to the pre-incubation NaOH treatment, which leads to fragmentation of semi-randomers (in the case of RNA and AtNA) and the generally lower RT-sensitivity for HNA and AtNA. (**C**) Size distributions of the RNA (teal) and AtNA (light red) recombination products. Compared to previous experiments (see *Figures 4* and *5*) RNA recombination products are biased towards shorter fragments, presumably due to increased hydrolysis during NaOH treatment. (**D**) Color-coded nucleotide signatures of AtNA recombination products from (**B**) shows similar bait segment truncation as in RNA (see *Figure 4*) but with a distinct ApN ligation signature.
DOI: https://doi.org/10.7554/eLife.43022.030

The following source data and figure supplements are available for figure 6:

**Source data 1.** Source data for panel C.
DOI: https://doi.org/10.7554/eLife.43022.037

**Figure supplement 1.** Addition of $Zn^{2+}$ does not promote ligation/recombination in semi-random sequence DNA or ANA pools while seeming to inhibit recombination in semi-random sequence RNA pools.
DOI: https://doi.org/10.7554/eLife.43022.031

**Figure supplement 1—source data 1.** Source data for panel A.
DOI: https://doi.org/10.7554/eLife.43022.032

**Figure supplement 2.** Susceptibility of RNA, HNA and AtNA oligomers to alkaline pH induced hydrolysis.
DOI: https://doi.org/10.7554/eLife.43022.033

**Figure supplement 3.** Proposed recombination mechanism of semi-random sequence AtNA pools.
DOI: https://doi.org/10.7554/eLife.43022.034

**Figure supplement 4.** Without alkaline pH (NaOH) treatment, recombination is only efficient in semi-random RNA pools.
DOI: https://doi.org/10.7554/eLife.43022.035

**Figure supplement 5.** Natural and synthetic polymers with vicinal diols (presumably) enabling ligation/recombination: ribonucleic acid (RNA), altritol-nucleic acid (AtNA), pyranoysl-RNA (pRNA) and (4'→3') or (4'→2') lyxopyranosyl-RNA (Lyxo). Genetic polymers to the left of dotted line (AtNA, Lyxo4', 3') can crosspair with natural nucleic acids, while polymers to the right (pRNA, Lyxo 4', 2') are orthogonal to the natural system.
DOI: https://doi.org/10.7554/eLife.43022.036

are designed to iteratively deplete the interaction network's underlying pool reactivity and instead isolate the exceedingly rare unitary sequences capable of function. While such experiments have been ground-breaking in revealing the potential of RNA (as well as DNA and some XNAs) for complex catalytic or ligand-binding functions (*Silverman, 2016*; *Taylor et al., 2015*; *Wilson and Szostak, 1999*), our results show that a back-extrapolation from the hit frequency of SELEX experiments leads to a gross underestimation of global pool functionality, as only single sequences are considered and systems-level interactions and networks are ignored.

We were able to visualize the intermolecular recombination and / or ligation products both directly by gel shift in denaturing gel electrophoresis and by specific RT-PCR amplification. Both assays provide evidence for *bona fide* ligation and recombination products as PCR or sequencing artefacts can be ruled out on multiple grounds. For one, incubation at $-80\ °C$, which freezes the eutectic phase in our buffer system (*Mutschler et al., 2015*; *Mutschler and Holliger, 2014*), never yields products, establishing that fragments do not recombine during workup. Furthermore, the appearance (or absence) of recombination products is clearly and reproducibly linked to chemical features of the original pool (e.g. free 5'OH (RNA) (*Figure 4*), or cis diol (RNA, AtNA) (*Figure 6*)), which are erased in the RT step. Finally, in both the case of the >p activated and unmodified random RNA sequence pools, we are able to observe the formation of ligation / recombination products directly by Urea-PAGE, which would dissociate even highly stable non-covalent complexes. Indeed, we note that our assays likely underestimate total pool reactivity (as well as the diversity of potential reaction products) as a range of more complex potential recombination modes, including intra- and intermolecular circular, lariat or branched products are only partially detected by our methods, or not at all (*Figure 4—figure supplement 4*).

Previous reports of non-enzymatic RNA recombination with defined substrates suggests that the majority of such reactions are likely to involve either a concerted mechanism of direct in-line attack on an internal phosphodiester linkage or, more frequently, a two step-mechanism of two consecutive transesterification reactions via an >p intermediate generated by hydrolysis (*Lutay et al., 2007*; *Nechaev et al., 2009*; *Staroseletz et al., 2018*). In both reaction trajectories, ligation /recombination is promoted by proximity through in trans intermolecular hybridization (*Figure 4—figure supplement 1*) as well as the concentration and entropic effects of eutectic ice phase formation. Whilst we observe formation of canonical 3'−5' linkages with some RNA motifs (H4), we expect that the majority of RNA pool linkages likely conform to 2'−5' regiochemistry as these linkages form preferentially in RNA duplex geometry (*Lutay et al., 2006*; *Orgel, 2004*), although such information is lacking for AtNA. In RNA, sporadic 2'−5' linkages have been found to be broadly compatible with folding and function (*Engelhart et al., 2013*) and can progressively be converted to the canonical 3'−5' linkage (*Mariani and Sutherland, 2017*).

The mechanistic basis for the distinctive sequence signature (CpN) in RNA is currently unclear, although CpA and UpA junctions have been shown to be generally more reactive (*Kaukinen et al., 2002*). Thus, the bias may reflect the higher propensity for C>p formation and/or reactivity, which may be more reactive due to the stronger electron withdrawal by the cytosine ring system (*Krishnamurthy, 2012*) or hydrogen bond interactions between the C-2 oxygen of the pyrimidine and the 2'-hydroxyl of the ribose (*Bibillo et al., 2000*). Earlier reports suggest that reactivity of phosphodiester bonds in a gapped duplex can be context dependent being strongly modulated by the local sequence (stacking) and base-pairing (H-bonding) (*Kaukinen et al., 2002*; *Oivanen et al., 1998*), which may explain the different AtNA recombination signature (ApN) (*Figure 6D*).

Repertoire selection experiments had previously shown that random sequence pools of different genetic polymers (including RNA, DNA and some XNAs) all contain functional sequences, including highly active ribozymes (*Wachowius et al., 2017*), DNAzymes (*Silverman, 2016*) and to a lesser extent ANA-, HNA- and other XNAzymes (*Taylor et al., 2015*). Although rare, these functional sequences can be isolated by iterative rounds of selection and amplification. In contrast, in the experiments presented here (which do not involve enrichment), we observe striking differences in innate reactive potential of different chemistries, with only RNA (and to a lesser extent AtNA) pools undergoing ligation and recombination. In contrast, DNA, ANA and HNA pools remained inert in our experiments, even under forcing conditions such as alkaline pH or high concentrations of transition metal ions like $Zn^{2+}$, which have been shown to promote self-cleavage in DNA (*Gu et al., 2013*). While nucleophilicity and $pK_a$ of the 5'-OH are likely to be comparable in RNA, DNA and the XNAs examined herein, classic studies using model nucleotide phosphodiester and -triester compounds

suggest an at least 30-fold higher reactivity of RNA internucleotide diester linkages compared to DNA and ANA (*Lönnberg and Korhonen, 2005*). This increased reactivity is thought to be due to the vicinal cis-diol configuration of RNA, with the 2′-OH stabilizing the negative charge developing on the phosphorane intermediate and /or the departing 3′-oxygen by both H-bonding (*Lönnberg and Korhonen, 2005*; *Oivanen et al., 1998*) and local electronic effects (*Aström et al., 2004*). Consistent with this mechanistic model, AtNA, an RNA congener, sharing an analogous vicinal cis-diol configuration (in a six-membered ring context), was found to be capable of intermolecular recombination, although AtNA reactivity appears to be significantly lower than RNA. The latter may be due to greater torsional strain required for in-line attack or formation of an analogous cyclic phosphate (AtNA>p) intermediate (with an almost planar configuration) in the context of a six-membered ring (*Bruzik et al., 1996*).

Although >p is only weakly activated towards nucleophilic attack, in the context of high effective concentrations of a 5′OH nucleophile in a gapped duplex (or pre-organized internal loop motifs like H4, J4), intermolecular bond formation can be efficient (*Usher and McHale, 1976*; *Lutay et al., 2006*). Crucially, >p groups form spontaneously as part of RNA (and presumably AtNA) hydrolytic degradation. While DNA pools have been found to comprise a number of small self-cleaving motifs (*Gu et al., 2013*), these yield 3′OH and 5′p termini, which require further activation for ligation. Thus a vicinal diol configuration not only accelerates the initial strand cleavage step of recombination, but also yields cleavage product termini (>p and 5′OH) competent for re-ligation. Indeed, other vicinal cis-diol containing genetic polymers have previously been examined by solid-phase synthetic chemistry as part of systematic variation along the ribo- and lyxopyranosyl series (*Eschenmoser, 1999*; *Wippo et al., 2001*) (*Figure 6—figure supplement 5*), and ligative polymerization of 2′, 3′>p activated pyranosyl tetranucleotides on a template has been demonstrated (*Bolli et al., 1997*). Therefore, a vicinal diol configuration appears to be a critical component of a spontaneous capacity for recombination, setting RNA apart from a range of other prebiotically plausible genetic polymers including TNA (*Schöning et al., 2000*), ANA (*Roberts et al., 2018*) and PNA (*Böhler et al., 1995*).

The pervasive spontaneous ligation and recombination activity displayed by oligomer pools described herein may seem counterintuitive, but such behaviour is both predicted and supported by theoretical arguments from statistical mechanics and thermodynamics. These suggest that under the right boundary conditions, recombination - with an concomitant increase in the diversity of length distribution of (RNA) polymer chains - is favoured by entropic factors (*Blokhuis and Lacoste, 2017*). Remarkably, not only is the diversity in oligomer length distribution altered but recombined pools also display a shift towards an increased tendency for secondary structure formation (*Figure 5E*). This observation likely reflects both the tendency of longer oligomers to form more stable and more diverse secondary structures (*Briones et al., 2009*; *Stich et al., 2008*) as well as some degree of intrapool selection for sequences capable of intermolecular hybridization and duplex formation (*Figure 1—figure supplement 2*). Finally, recombination also scrambles parts of the pool sequence information, which could lead to an increase in informational complexity in redundant pools such as the random 20-mer RNA and semi-random 40-mer RNA and AtNA oligomer pools used in our experiments. To better understand this effect, we explored recombination outcomes by in silico simulations of a broad range of boundary conditions (including a range of hydrolytic cleavage (degradation) vs. ligation rates, pool redundancy and code complexity (2-letter vs. 4-letter code)). Using the Shannon population level entropy index D (*Derr et al., 2012*) as an established measure of informational complexity (see Material and methods), we find that recombination in simulated, redundant $N_{10}$ and $N_{20}$ pools disproportionates pool sequences into a distribution of both shorter (cleavage) and longer (ligation) strands and spontaneously increases D under a wide range of boundary conditions (*Figure 7*, *Figure 7—figure supplement 1*). Furthermore, in silico recombination reveals critical pool and reaction parameters that enhance information gains, whereby increases in pool informational complexity (as measured by *D*) scale with pool redundancy, code complexity and oligomer length (*Figure 7—figure supplement 1*). Under the conditions used in our experiments, i.e. ca. 1000-fold pool redundancy, a 4-letter base alphabet and $N_{20}$ oligomer length, information gains occur under almost all boundary conditions (*Figure 7*). Thus, spontaneous recombination in random oligomer pools under any but the most strongly degradative regimes expands both sequence size distribution and sequence diversity, and thereby progressively increases pool compositional (sequence length distribution), structural (secondary structure formation) and informational complexity.

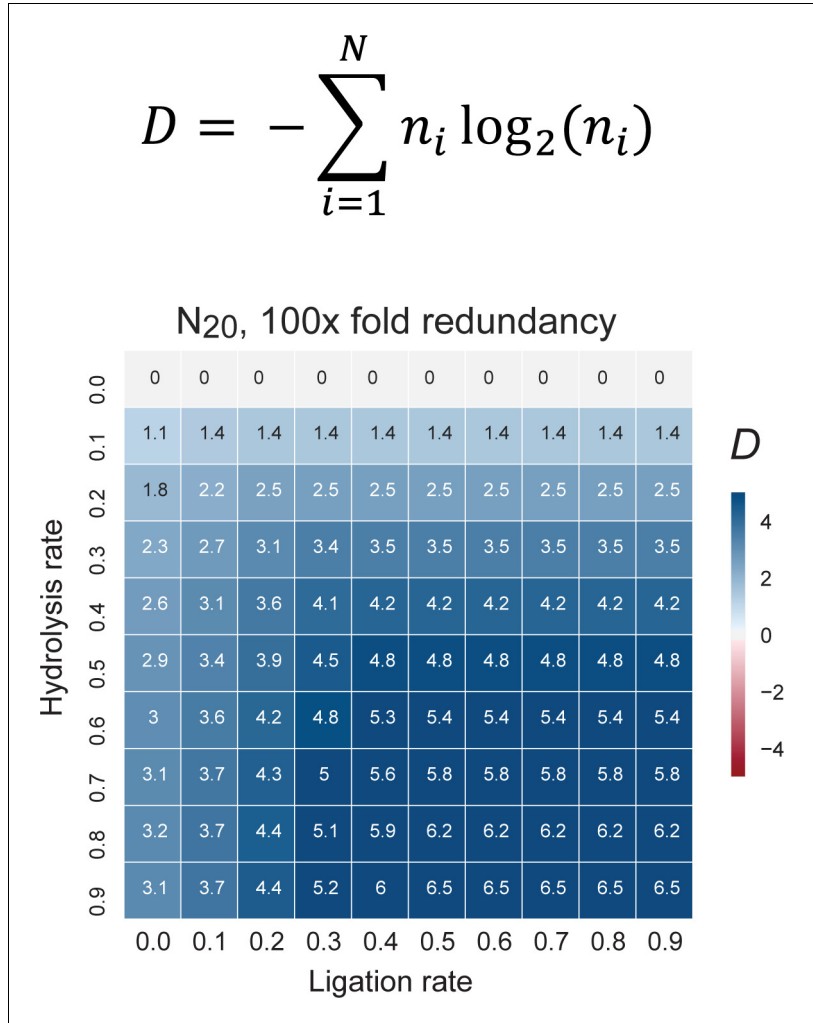

**Figure 7.** The change in information content in a random sequence pool as a consequence of recombination. Informational content / complexity change is computed as the population level Shannon entropy (Diversity index: $D$ (**Derr et al., 2012**)), where $N$ is the total number of unique sequences in the pool, and $n_i$ is the proportion of individuals belonging to the $i^{th}$ unique sequence. $D$ is zero if all molecules are identical and $D$ is maximal when molecules are uniformly distributed through sequence space (top panel). $D$ changes after recombination of a $N_{20}$ 4-letter code, random sequence pool of 100-fold redundancy (each sequence is present in 100 copies). The difference of pool diversity before and after recombination ($\Delta D$) plotted as a heat map as a function of hydrolysis and ligation rates (bottom panel). Under these boundary conditions (mimicking the RNA pools experimentally analysed for example in **Figure 5**), $D$ increases under all examined combinations of hydrolysis and ligation rates. DOI: https://doi.org/10.7554/eLife.43022.038

The following figure supplement is available for figure 7:

**Figure supplement 1.** The change in information content in a random sequence pool as a consequence of recombination under different boundary conditions. DOI: https://doi.org/10.7554/eLife.43022.039

In the context of evolution of early random sequence oligomer pools (and potentially viral quasispecies), the impact of the above recombination processes is significant, in particular considering that such pools would at the same time likely to be under fractionation and/or adaptive pressure. Indeed, a number of physicochemical processes (including thermophoretic cycles in porous substrates (**Kreysing et al., 2015**) or adsorption to mineral surfaces (**Ferris et al., 1996**)) would favour length-extended oligomers over pool members or short cleavage products. However, adaptive pressure is arguably the most important process that favours more diverse and length extended oligomer

sequences. Previous studies have underlined the importance of oligomer length for encoding the most active (large) functional motifs or more frequent smaller motifs (*Ekland et al., 1995*; *Velez et al., 2012*; *Bartel and Szostak, 1993*), as well as the importance of sequence diversity and the power of (continuous) recombination in accelerating adaptive walks (*Stemmer, 1994*; *Hayden et al., 2005*; *Lehman, 2008*; *Arnold, 2009*; *Mutschler et al., 2015*).

In conclusion, our findings indicate that native random-sequence RNA (and to a lesser extent AtNA) pools are naturally predisposed to recombination via near energetically neutral transesterification chemistry enabled by their specific chemical structure. As a result, such pools have an unique innate capacity to 'bootstrap' themselves towards higher compositional, informational and structural complexity (*Briones et al., 2009*) and thus a likely enhanced adaptive potential.

## Materials and methods

### Oligonucleotides

Oligonucleotides are described in detail in *Supplementary file 2* (RNA) and *Supplementary file 3* (DNA). RNAs with terminal 2′, 3′-cyclic phosphate were created from 2′/3′-phosphorylated RNAs using 1-ethyl-3-(3-dimethylaminopropyl)carbodiimide (EDC, Sigma) as described previously. (*Mutschler et al., 2015*; *Mutschler and Holliger, 2014*) Non-random activated RNAs were further purified by Urea-PAGE. Random EDC-treated RNAs were precipitated with isopropanol and washed extensively with 70% ethanol. Dried RNA pellets (3–8 nmol RNA) were resuspended in nuclease-free $H_2O$ and stored at $-30$ °C until usage. All oligonucleotide concentrations were determined spectroscopically using the extinction coefficients at 260 nm calculated by OligoCalc (*Kibbe, 2007*). To approximate the concentration of random RNAs, an even nucleotide distribution was assumed. Random XNA pools were prepared using DNA primers and templates described in *Supplementary file 2* and polymerase D4K (*Pinheiro et al., 2012*) for ANA, or 6G12 I521L (*Taylor et al., 2015*) for HNA as described previously (*Pinheiro et al., 2012*; *Taylor et al., 2015*; *Taylor and Holliger, 2015*). For AtNA synthesis, we developed a novel polymerase, polLG, which will be described elsewhere. Following XNA synthesis, DNA components were degraded using 0.1 U/µl TURBO DNase (Invitrogen) for 2 hr at 37 °C in 10 mM Tris•HCl (pH 7.6), 2.5 mM $MgCl_2$ and 0.5 mM $CaCl_2$, and all-XNA oligonucleotides purified by Urea-PAGE.

### Ligation / recombination reactions

For semi-random bait/prey RNA pairs (*Supplementary file 2*), 2.5 µM of both sets of fragments were frozen on dry ice or at $-30$ °C in 25 mM NaCl, 1 mM Tris•HCl pH 8.3 and transferred in a TXF200-R3 refrigerated water-baths (Grant) filled with 50% (v/v) Blue Star Antifreeze (Carplan) equilibrated at $-9$ °C. Fully random oligonucleotides pools were frozen at a higher initial concentration of 16 µM ($N_{20}$>p) supplemented with 1 µM of FITC-labelled $N_{20}$>p pool. For analytical experiments with defined RNA sequences (H4, J4, splint samples, minimal hairpin constructs) 0.5 µM of a FITC-labelled 3′-fragment was mixed with 1 µM of non-fluorescent, >p activated 5′-ligation partner. For analytical *trans* reactions, RNA splints concentrations were 1 µM. For preparative-scale reactions, concentrations were increased 10-fold to improve ligation efficiencies and product yields. Naive FITC-$N_{20}$ / $N_{20}A_{10}$ pools (2.5 µM each) were incubated at for 5 months before recovery and denaturing PAGE analysis. Partially hydrolysed semi-random RNA and AltNA bait/prey pairs (see below) were incubated at a final concentration of 0.6 µM for each oligonucleotide for 5 months at $-9$ °C.

All samples, with the exception of those used for qRT-PCR and deep-sequencing, were quenched with an excess of 4x quench buffer (98% formamide, 10 mM EDTA pH 8, 0.04 wt% bromophenol blue) then analysed by denaturing PAGE using 20% (w/v) Acrylamide/Bis 19:1 TB/8 M urea gels. For ligation kinetics, reaction master mixes were aliquoted into 10 µl reactions and frozen/quenched independently to prevent repeated freeze-thaw cycles of individual reactions. Gels were imaged using a Typhoon Trio scanner (GE Healthcare). Percentage intensity of starting material and ligation products were determined using ImageQuant TL. For quantifications of SYBRgold-stained $N_{20}$>p ligation products, band intensities were normalized using the expected oligonucleotide length (*Supplementary file 1*) before calculating the relative ligation yields. Gel image brightness and contrast were adjusted to illustrate ligation product banding patterns. PAGE gels do not show any bands other than those presented in the figures. Samples used for qRT-PCR and deep sequencing

were stored at −80 °C until needed. cDNA synthesis and library preparation from $N_{20}$>p ligation reactions.

Ice pellets of preparative ligation reactions (1.6 nmol of total $N_{20}$>p in 100 µl) from −9 °C long-term incubation experiments were thawed on ice. 5 µl were mixed with 4 volumes of denaturing gel loading buffer (95% formamide, 10 mM EDTA pH 8, 0.1% (w/v) bromophenol blue) and analysed by denaturing PAGE followed by fluorescent gel imaging. To phosphorylate free 5'-ends and dephosphorylate unreacted 3'-ends, the remaining 95 µl of the reaction was treated with 90 U T4-PNK (NEB) for 45 min at 37 °C in presence of 1 mM ATP and 1x T4-PNK reaction buffer. 10 pmol of fresh $N_{20}$>p was treated the same way to obtain sequence information of the starting material. T4-PNK was quenched by the addition of 8.8 µmol EDTA and formamide (final concentration 40%). For in-ice ligation reaction, ligated products were separated from the $N_{20}$>p starting material by preparative denaturing PAGE followed by staining with SYBR Gold RNA stain (ThermoFisher Scientific). The bands corresponding to the $N_{40}$ ligation product were excised, eluted from the gel matrix in 0.3 M sodium acetate (pH 5) and precipitated using 2 volumes of isopropanol. To remove residual gel-contaminations, both RNA samples were further purified using the RNeasy Mini Kit (Qiagen). Next, 3.5 pmol of the RNA pools (or the whole 10 pmol of $N_{20}$>p starting material) were ligated to preadenylated adapter Appa-3RND-Cy5 (20 pmol, **Supplementary file 3**) using 200 U T4 RNA Ligase 2, truncated KQ (NEB) in presence of 15% (w/v) PEG 8000 and RNA ligase buffer (1x) for 90 min at 25 °C. After addition of 22 pmol 3RND_rev primer in 5.5 µl $H_2O$, the sample was incubated for 5 min at 72 °C, 15 min at 37 °C and 15 min at 25 °C followed by addition of 14.5 µl 5' linker ligation mix (4 µl of 10 mM ATP, 2 µl of 10x T4 RNA Ligase buffer (NEB), 5.5 µl of 50% PEG 8000, 1 µl of 20 µM 5_RNDM linker and 2 µl RNA Ligase 1 (NEB, 10 U /µl). The ligation mix was incubated for 2 hr at 25 °C and stored at −20 °C. RT-PCR (50 µl) was performed using 1 µl of the ligation reaction and 20 pmol of the recovery primers 5RNDM_fw/3RNDM_rv in a SuperScript III One-Step RT-PCR reaction supplemented with Platinum Taq DNA Polymerase (ThermoFisher Scientific) according to the supplier's instructions. Amplified cDNA samples were prepared for Illumina deep sequencing by appending the bridge-amplification sequences by PCR using the NEBNext High-Fidelity 2X PCR Master Mix (NEB) using the primers P3_3RND and P5_5RND according to the supplier's protocol. Libraries were barcoded using P5 primers containing 6 nt sequences from the NEXTflex series (Illumina).

## cDNA synthesis from semi-random RNA reactions

For semi-randomer RNA reactions, RT-PCR was performed using the SuperScript III One-Step RT-PCR System with Platinum Taq DNA Polymerase (Invitrogen), according to the manufacturer's protocol. In each reaction, 1.25 pM of total RNA from ligation or control reactions was used as RT-template with forward and reverse primers complementary to the respective constant $C1_{20}/C3_{20}$ and $C2_{20}/C4_{20}$ primer binding sites (**Supplementary file 3**). Deep sequencing libraries were prepared using the respective P5_C1/P5_C3 and P3_C2/P3_C4 primers using the NEBNext High-Fidelity 2X PCR Master Mix (NEB). To prepare $C1_{20}/C3_{20}$-$N_{20}$>p pre-ligation for sequencing, the >p group was removed by dephosphorylation using T4-Polynucleotide Kinase (NEB) followed by adapter ligation using miRNA Cloning Linker 1 (IDT) as described in the previous section. cDNA was generated by RT-PCR using primers C1_fw (or C3_fw) and miRNA_CL1_rev in a SuperScript III One-Step RT-PCR reaction supplemented with Platinum Taq DNA Polymerase (ThermoFisher Scientific) according to the supplier's instructions. Libraries for Illumina deep sequencing were prepared by PCR using the NEBNext High-Fidelity 2X PCR Master Mix (NEB) using the primers P3_miRNA_CL1_rv and P5_C1 (or P5_C3) according to the supplier's protocol. DNA libraries from the 5'OH-$N_{20}$-$C2_{20}$ and 5'OH-$N_{20}$-$C4_{20}$ pre-ligation semi-random RNA pools were generated using a two-step protocol: First, single stranded cDNA was prepared using C2_rv/C4_rv using the SuperScript III according to the supplier's instructions. Next, miRNA Cloning Linker 1 was ligated to the purified cDNA and ds cDNA was generated using GoTaq Green mastermix (Promega). Finally, reverse complement cDNA libraries of the 5'OH-$N_{20}$-$C2_{20}$ and 5'OH-$N_{20}$-$C4_{20}$ were generated using P3_miRNA_CL1_rv and P5_C2 (or P5_C3).

## cDNA synthesis from $N_{20}A_{10}$ / $N_{20}$ recombination reactions

For selected reactions with fully random pools, polyU tails were added to provide a primer-binding site prior to RT-PCR. PolyU tailing was performed using 0.2 U/µl polyU polymerase (NEB) in 1X NEB

Buffer 2, 0.5 mM UTP for 20 min at 37 °C, followed by heat inactivation for 1 min at 75 °C. Next, RT-PCR was performed (with or without prior polyU-tailing) using the SMARTer RT-PCR system (Clontech), according to the manufacturer's protocol, using the supplied SMARTer oligo as forward primer, and an oligo containing either a polyA- or polyT as reverse primer (*Supplementary file 3*; *Figure 5—figure supplement 1*). cDNA was amplified using OneTaq hot-start master mix (NEB) supplemented with appropriate primers (*Supplementary file 3*).

## DNA synthesis from semi-random XNA reactions.

XNA reverse transcription was primed using DNA or mixed DNA-2'-O-methyl-RNA primers (*Supplementary file 3*). ANA was reverse transcribed using RTI521L as described previously (*Pinheiro et al., 2012*; *Taylor and Holliger, 2015*). HNA and AltNA were reverse transcribed using a novel polymerase, polTK[2], which will be described elsewhere. XNA RT reactions were isolated using streptavidin beads as described previously (*Taylor and Holliger, 2015*), prior to amplification using OneTaq hot-start master mix (NEB), or a blend of OneTaq and ThermoScript (Thermo Fisher), with appropriate primers (*Supplementary file 3*).

## qRT-PCR

qRT-PCR was performed according to the supplier protocol for SuperScript III One-Step RT-PCR System with Platinum Taq DNA Polymerase in a DNA Engine Opticon System 2 (Bio-Rad). 1.25 pmol RNA was used with 6 pmol C1_fwd and C2_rev primers in 30 μl reaction volume.

## Deep sequencing and pool analysis

Amplified polyclonal cDNA samples were prepared for deep sequencing by the Illumina Miseq method by appending the bridge-amplification sequences by PCR, as described previously (*Taylor and Holliger, 2015*). PCR products were purified using either a QIAquick PCR Purification kit (Qiagen), or by agarose gel followed by a QIAquick Gel Extraction kit (Qiagen). 6–12 pmol of pooled libraries plus 20% - 30% PhiX control (Illumina) were denatured and sequenced (single-end read, 75 or 150 cycles) using a MiSeq reagent kit and instrument (Illumina) according to manufacturer's instructions. Libraries were barcoded using P5 primers containing 6 nt sequences from the NEXTflex series (Illumina). Data files were trimmed, quality filtered (90%, Q20), and collapsed using scripts combining the FASTX command line toolkit (http://hannonlab.cshl.edu/fastx_toolkit/commandline.html) and cutadapt (*Martin, 2011*) on a Linux workstation. All FASTA files containing sequences of ligation and recombination products were split according to their product size using either *awk* or custom bash scripts implementing the *egrep* command (an example script is provided as *Source code 1*). Position-specific nucleotide frequencies were determined using the *fastx_quality_stats* command. To create nucleotide signature plots, these frequencies were converted into RGB code at each position $i$ using the following transformations:

$$R_i = 255 - [f_C i \cdot 255] - [f_A i \cdot 255]$$

$$G_i = 255 - [f_C i \cdot 255] - [f_U i \cdot 255]$$

$$B_i = 255 - [f_A i \cdot 255] - [f_G i \cdot 255] - [f_U i \cdot 255]$$

where $f_C i$, $f_A i$, $f_U i$, and $f_G i$ are the positional nucleotide frequencies for C, A, U and G at each position $i$ respectively with $f_C i + f_A i + f_U i + f_G i = 1$.

To determine ligation/recombination-specific secondary structure (ss) biases in the experimental data sets, large (>100,000 sequences) datasets were generated with a custom Python script (*Source code 2*) using the experimental position-specific nucleotide frequencies as input. Minimal free energy (MFE) structures for both, experimental and synthetic datasets were calculated using *RNAfold* (*Lorenz et al., 2011*) using standard settings. The position-specific dot-bracket element frequencies were extracted using custom bash scripts (an example script is provided as *Source code 3*).

To screen for secondary structures promoting ligation, the minimal free energy ss for each sequence from a C3$_{20}$-N$_{20}$>p × 5'OH-N$_{20}$-C2$_{20}$ and C1$_{20}$-N$_{20}$>p × 5'OH-N$_{20}$-C4$_{20}$ ligation experiment was calculated using *RNAfold* 2.3.5 with default settings. Using custom bash scripts (example script provided as *Source code 4*), all position-specific matches between ss submotifs from

*Figure 2—figure supplement 3* and the calculated ss in proximity to the ligation site were clustered. Nucleotide sequences of matched ss-motifs were extracted into separate FASTA-files and consensus motifs for selected matches with untypically increased ligation frequencies in unpaired regions were determined using DREME (*Bailey, 2011*). Motif H4 and J4 gave the most unambiguous nucleotide consensus sequences and were therefore experimentally tested for ligation.

## Regiospecificity of H4

3′−5′-linked controls of full-length H4 or H4T1 were produced in 75 µl reactions containing 5 nmol of 5H4 or 5H4T1 and 3H4 or 3H4T1 RNA in T4 DNA Ligase Buffer (NEB). In a first step, the 3′-fragment was 5′ phosphorylated and the 5′-fragment dephosphorylated using 30 U of full-length T4 Polynucleotide Kinase (NEB) followed by incubation for 1 hr at 37 ˚C. After addition of an equimolar amount of H4_splintG (for H4T1) the reaction was heated to 80 ˚C for 1 min and transferred to 65 ˚C for 15 min. The sample volume was made up to 100 µl with 2.5 µl 10x T4 DNA Ligase Buffer, $H_2O$ and 30 U T4 RNA Ligase 2 (NEB). No splint was necessary for enzymatic ligation of H4. Samples were incubated at 37 ˚C for 90 min followed by gel purification of the ligated RNA. Self-ligated H4 RNA was obtained by gel purification of the 80 nt band after large-scale self-ligation of 5H4>p and 3H4 at −9 ˚C in 25 mM NaCl, 1 mM Tris•HCl pH 8 in a TXF200-R3 refrigerated water-baths for 2 weeks. Similarly, self-ligated H4T was produced by in-ice ligation 5H4T>p and 3H4T in presence of equimolar amounts of H4_splintA.

For analytical DNAzyme cleavage, 0.1 µM ligated H4 sample or control was mixed with 1 µM E5112_H4 designed to cleave the H4 GrA ligation junction (*Cruz et al., 2004*) in E5112-DNAzyme buffer (250 mM KCl, 400 mM NaCl, 125 mM HEPES•KOH pH 7, 50 mM $MgCl_2$, 18.75 mM $MnCl_2$) and incubated for 2 h hours at 37 ˚C. Reactions were quenched by adding 4 volumes quenching buffer (98% formamide, 10 mM EDTA, 0.1% (w/v) Bromophenol Blue) and analysed by Urea-PAGE followed by fluorescent imaging. For RNaseT1 cleavage, 0.15 µM ligated H4T1 was incubated at 55 ˚C in 25 µl RNase T1 Buffer (30 mM sodium acetate pH 5, 1 µM EDTA, 30 mM urea, 0.04 U/µl RNaseT1). Samples were quenched at 0, 10 and 30 min using 4 volumes of quenching buffer and analysed using Urea-PAGE followed by fluorescence gel imaging.

## Regioselectivity of J4

An all 3′−5′-linked control for J4-shrt was generated by annealing 5 nmol of each J4_shrt-P and J4_shrt-F with 5 nmol of the DNA splint J4_shrt-splint followed by ligation using T4-RNA Ligase two as described above. Self-ligated J4-shrt product was generated by co-incubating J4_shrt>p and J4_shrt-F in 25 mM NaCl, 1 mM Tris•HCl pH 8 in a TXF200-R3 refrigerated water-baths equilibrated at −9 ˚C for 2 weeks. Both self-ligated and control ligation products were gel-purified. Self-cleavage of J4-shrt RNAs was probed in 25 mM HEPES•NaOH pH 7.5, 5 mM $MgCl_2$ and 5 mM $MnCl_2$ (room temperature, 15 h). Cleavage of the 3′−5′ J4-shrt control at the J4 ligation junction was performed using the custom 8–17-derived DNAzyme under the same conditions. The same DNAzyme was also used to inhibit initial self-cleavage of in-ice self-ligated J4.

## Partial hydrolysis of random XNA and RNA pools

For experiments in which random oligonucleotide pools were partially hydrolysed prior to ligation reactions, purified 40mer oligonucleotides (equivalent to C1-bait, *Supplementary file 2*) were incubated in 50 mM NaOH at 65 ˚C for 3 min (RNA) or 27 min (AtNA and HNA), neutralized with 1 M Tris•HCl (pH 7.4).

## Altritol triphosphates

Altritol nucleosides (*Abramov and Herdewijn, 2007*) were converted to their 5′-triphosphates in one-pot reaction by Lugwig method (*Ludwig, 1981*). In this procedure, regioselective phosphorylation of 5′-hydroxy group of sugar was carried out with phosphoryl oxychloride in trimethylphosphate followed by the addition of tetrabutylammonium pyrophosphate. The triphosphates were deprotected with ammonia, isolated with ion-exchange chromatography and finally purified by RP HPLC.

**Scheme 1.** Nu = A$^{Bz}$, C$^{Bz}$, $^{dmf}$G, and U. **i)** POCl$_3$, trimethylphosphate; **ii)** tetrabytylammonium pyrophosphate, NBu$_3$, DMF; **iii)** 25% NH$_3$.
DOI: https://doi.org/10.7554/eLife.43022.040

For all reactions, analytical grade solvents were used. All moisture-sensitive reactions were carried out in oven-dried glassware under N$_2$. Reagents and solvents were provided by Acros, Fluka, or TCI. TLC: $^{31}$P NMR spectra: a Bruker Avance 300-MHz, or a Bruker Avance 500-MHz spectrometer. HRMS spectra were acquire on a quadrupole orthogonal acceleration time-of-flight mass spectrometer (Synapt G2 HDMS, Waters, Milford, MA). Samples were infused at 3 µl/min and spectra were obtained in positive (or: negative) ionization mode with a resolution of 15000 (FWHM) using leucine enkephalin as lock mass.

General procedure for the synthesis of altritol nucleoside 5' triphosphates: To an ice-cold solution or suspension of corresponding altritol nucleoside (*Abramov and Herdewijn, 2007*) (0.1 mmol) in trimethyl phosphate (TMP) (1.0 mL) was added phosphoryl chloride (20 µl, 0.22 mmol) and the solution was stirred at 0 °C for 5 hr. Tributylamine (300 µl, 1.6 mmol) and tetrabutylammonium pyrophosphate solution (0.5 M in DMF, 1.1 mmol) were added simultaneously, and the solution was stirred for a further 30 min. The reaction was then quenched by the addition of 0.5 M triethylammonium bicarbonate (TEAB) buffer (10 mL), and stored at 4 °C overnight. The reaction product was deprotected with 25% ammonia, evaporated to dryness, re-dissolved in 0.1 M TEAB (5 mL) and applied to a Sephadex A25 column. The column was eluted with a linear gradient of 0.1–1.0 M TEAB. Appropriate fractions were pooled and evaporated to dryness to give desired product. The triphosphates were finally purified by HPLC (Alltima 5µ C-18 reverse phase column 10 × 250 mm, buffer A, 0.1 M TEAB; buffer B, 0.1 M TEAB, 25% MeCN. 0% to 100% buffer B over 60 min at 4 mL / min).

1′,5′-ANHYDRO-2′-DEOXY-2′-(ADENIN-9-YL)-D-ALTRITOL 6′-TRIPHOSPHATE TRIETHYLA MMONIUM SALT (*Vastmans et al., 2001*). $^{31}$P NMR (D$_2$O): δ −9.2 (1P, d), −11.1 (1P,d), −23.0 (1P, t). ESIHRMS found: (M-H)$^-$ 520.0044. calcd for C$_{11}$H$_{18}$N$_5$O$_{13}$P$_3$: (M-H)$^-$ 520.0041.

1′,5′-ANHYDRO-2′-DEOXY-2′-(GUANIN-9-YL)-D-ALTRITOL 6′-TRIPHOSPHATE TRIETHYLA MMONIUM SALT $^{31}$P NMR (D$_2$O): δ −8.4 (1P, br s), −10.9 (1P, d), −22.5 (1P, t). ESIHRMS found: (M-H)$^-$ 535.9991. calcd for C$_{11}$H$_{18}$N$_5$O$_{14}$P$_3$: (M-H)$^-$5359990.

1′,5′-ANHYDRO-2′-DEOXY-2′-(CYTOSIN −1-YL)-D-ALTRITOL 6′-TRIPHOSPHATE, TRIETHYLA MMONIUM SALT $^{31}$P NMR (D$_2$O): δ −6.9 (1P, br d), −11.0 (1P,d), −22.7 (1P,br t). ESIHRMS found: (M-H)$^-$ 495.9938. calcd for C$_{10}$H$_{18}$N$_3$O$_{14}$P$_3$: (M-H)$^-$ 495.9928.

1′,5′-ANHYDRO-2′-DEOXY-2′-(URACIL-1-YL)-D-ALTRITOL 6′-TRIPHOSPHATE, TRIETHYLA MMONIUM SALT $^{31}$P NMR (D$_2$O): δ −6.5 (1P, d), −11.1 (1P,d), −22.6 (1P,t). ESIHRMS found: (M-H)$^-$ 496.9795. calcd for C$_{10}$H$_{17}$N$_2$O$_{15}$P$_3$: (M-H)$^-$ 496.9768

## Measuring population-level entropy

To quantify the informational content, *D,* of a population of sequences, we compute the Shannon index (*Derr et al., 2012*):

$$D = -\sum_{i=1}^{N} n_i \log_2(n_i)$$

Where *N* is the total number of unique sequences in the pool, and $n_i$ is the proportion of individuals belonging to the $i^{th}$ unique sequence. *D* is zero if all molecules are identical and *D* is maximal when molecules are uniformly distributed through sequence space.

### Simulations of recombination reactions

To understand the effect of recombination on informational content, we simulated a population of random sequences that can undergo hydrolysis according to a normal distribution around the midpoint of the sequence in question. Hydrolysed sequences of length greater than two nucleotides are

then allowed to randomly ligate with either another hydrolysed sequence, or an unhydrolysed sequence, weighted by the size of each population. The Shannon index is computed for the population before and after recombination, and the delta Shannon index, $\Delta D$, is reported. Here, a negative $\Delta D$ indicates a loss in informational content and a positive $\Delta D$ indicates a gain of informational content. The Python script for the simulation is available as *Source code 5*.

## Acknowledgments

The authors thank J Attwater for discussions and comments on the manuscript.

This work was supported by a Federation of European Biochemical Societies (FEBS) Long-Term Fellowship (HM), the Medical Research Council (PhH, AIT, GH, program no. MC_U105178804) and from KU Leuven, Research Council (OT/1414/128) and FWO Vlaanderen (G078014N) (MA and PiH).

## Additional information

### Funding

| Funder | Grant reference number | Author |
|---|---|---|
| Medical Research Council | MC_U105178804 | Alexander I Taylor<br>Benjamin T Porebski<br>Gillian Houlihan<br>Philipp Holliger |
| Federation of European Biochemical Societies | FEBS Long-Term Fellowship | Hannes Mutschler |
| KU Leuven | OT/1414/128 | Mikhail Abramov<br>Piet Herdewijn |
| Fonds voor Wetenschappelijk Onderzoek - Vlaanderen | G078014N | Mikhail Abramov<br>Piet Herdewijn |

The funders had no role in study design, data collection and interpretation, or the decision to submit the work for publication.

### Author contributions

Hannes Mutschler, Conceptualization, Resources, Software, Formal analysis, Supervision, Validation, Investigation, Visualization, Methodology, Writing—original draft, Project administration, Writing—review and editing; Alexander I Taylor, Conceptualization, Resources, Formal analysis, Validation, Investigation, Visualization, Methodology, Writing—original draft, Writing—review and editing; Benjamin T Porebski, Software, Validation, Visualization, Methodology; Alice Lightowlers, Validation, Investigation; Gillian Houlihan, Resources, Developed, produced and tested the new RT-enzyme; Mikhail Abramov, Resources, Synthesized and quality-checked the HNA and AtNA nucleosides used in this study; Piet Herdewijn, Resources, Funding acquisition; Philipp Holliger, Conceptualization, Resources, Formal analysis, Supervision, Funding acquisition, Validation, Visualization, Methodology, Writing—original draft, Project administration, Writing—review and editing

### Author ORCIDs

Hannes Mutschler (iD) http://orcid.org/0000-0001-8005-1657
Alexander I Taylor (iD) http://orcid.org/0000-0001-7684-1437
Piet Herdewijn (iD) http://orcid.org/0000-0003-3589-8503
Philipp Holliger (iD) http://orcid.org/0000-0002-3440-9854

### Decision letter and Author response

Decision letter https://doi.org/10.7554/eLife.43022.053
Author response https://doi.org/10.7554/eLife.43022.054

## Additional files

### Supplementary files

• Source Code 1. Example shell script for the extraction of the sequence length distributions from a FASTA file.
DOI: https://doi.org/10.7554/eLife.43022.041

• Source Code 2. Python script for the generation of FASTA files containing random DNA sequences based on a given nucleotide distribution. The script requires a tab-separated ASCII-file containing tab-separated frequencies (0–1) of the bases A,C,G,T,N as input.
DOI: https://doi.org/10.7554/eLife.43022.042

• Source Code 3. Shell script for the extraction of dot-bracket frequencies. Requires RNAfold and a FASTA-file as input.
DOI: https://doi.org/10.7554/eLife.43022.043

• Source Code 4. Shell script used to identify frequent ligation motifs using an RNAfold file as input.
DOI: https://doi.org/10.7554/eLife.43022.044

• Source Code 5. Python script for the simulated recombination reactions and Shannon Index calculations.
DOI: https://doi.org/10.7554/eLife.43022.045

• Supplementary file 1. Quantification of $N_{20}>p$ ligation products (Urea-PAGE, stained with SYBRgold).
DOI: https://doi.org/10.7554/eLife.43022.046

• Supplementary file 2. RNA oligonucleotides used in this study.
DOI: https://doi.org/10.7554/eLife.43022.047

• Supplementary file 3. DNA oligonucleotides used in this study.
DOI: https://doi.org/10.7554/eLife.43022.048

• Transparent reporting form
DOI: https://doi.org/10.7554/eLife.43022.049

### Data availability

All data generated or analysed during this study are included in the manuscript and supporting files. Source data files have been provided for: Figure 1, Figure 1—figure supplement 1, Figure 2, Figure 2—figure supplement 1, Figure 4, Figure 4—figure supplement 2, Figure 4—figure supplement 3, Figure 5, Figure 5—figure supplement 1, Figure 6 and Figure 6—figure supplement 2. Further source files are shell scripts for the motif search, extraction of secondary structure frequencies and size distributions and python scripts for the generation of random sequences using position-specific nucleotide frequencies and the simulations of the population level Shannon Index at different cleavage / ligation rates are available online (see Source code 1-5).

The following dataset was generated:

| Author(s) | Year | Dataset title | Dataset URL | Database and Identifier |
|---|---|---|---|---|
| Mutschler H, Taylor AI, Porebski BT, Lightowlers A, Houlihan G, Abramov M, Herdewijn P, Holliger P | 2018 | Data from: Random-sequence genetic oligomer pools display an innate potential for ligation and recombination | https://dx.doi.org/10.5061/dryad.gj8jv50 | Dryad Digital Repository, 10.5061/dryad.gj8jv50 |

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
