## [Decision Letter]

[Editors’ note: a previous version of this study was rejected after peer review, but the authors submitted for reconsideration. The first decision letter after peer review is shown below.]

Thank you for submitting your article "Innate potential of random genetic oligomer pools for recombination" for consideration by *eLife*. Your article has been evaluated by three peer reviewers, and the evaluation has been overseen by a guest Reviewing Editor and Detlef Weigel as Senior Editor.

The reviewers have discussed the reviews with one another. As you will see from the individual reviews, there was a range of opinions regarding the importance and validity of the work. Overall, the work was seen as technically strong, but there were major concerns about the impact of the work. All three reviewers and the guest editor agreed that very substantial revisions would be necessary before the manuscript could be considered again for *eLife*. Most importantly, reviewer 2 raises the point that the manuscript is too focused on the audience of nucleic acid chemists; a re-submitted manuscript would have to demonstrate, supported by literature, that the study is of interest to a wider audience. Otherwise, reviewer 2 and the guest editor feel that the manuscript would be appropriate for more specialized journals such as Nucleic Acids Research or Angewandte Chemie.

At this point we cannot directly invite revision, but we do remain interested in the work. We would likely reconsider the work, if you can make a much more convincing case regarding the general interest and advance of your findings, plus fully address the following concerns (and address the rest of the reviewers' comments as much as possible).

Essential revisions:

Reviewer 1: major comments 1 and 4.

Reviewer 2, major comment 1: please provide clear arguments, supported by literature, why this study is of interest to a broader audience than nucleic acid chemists, by inserting a section into the introduction. The first paragraph of reviewer 3 (last sentence) could be used as starting point. Major comment 2: Based on this concern, please describe in additional sentences how the reactions monitored in the study could give rise to an overall increase in compositional complexity if both cleavage and ligation events are considered.

Reviewer 3: all major comments.

*Reviewer #1:*

This study investigates the ability of random sequence pools of RNA, DNA, ANA, HNA, and AtNA to promote ligation and recombination reactions. The work is original, and explores important concepts related to the functional potential of polymers. However, from my perspective it could be significantly improved by considering the following issues.

1) You used gel purification and high-throughput sequencing to identify molecules from different types of random sequence pools that undergo ligation or recombination reactions after long incubations. My major criticism of this work is that you did not show that these sequences react at enhanced rates relative to the average polymer sequence. Each pool probably contains a small number of molecules with enhanced ligation or recombination rates and a large excess of sequences that react at background levels. After a single purification (analogous to a single round of selection), it is possible that these pools will still contain mostly inactive sequences. If this is the case, sequencing will not necessarily provide information about motifs that promote ligation/recombination (but see point two). The clearest way to address this point would be to experimentally measure the ligation/recombination rates of randomly chosen variants from at least one of the evolved pools. Are the ligation/recombination rates of these sequences higher than those of arbitrary sequences or the unevolved starting pool? Figure 2—figure supplement 4 shows that you can measure the rates of some of these sequences in a reasonable amount of time, but not whether they are enhanced relative to arbitrary sequences or the starting pool. The fact that the sequence compositions of the ligated/recombined pools are different from the starting pools is encouraging in this respect. On the other hand, because the results of many experiments in this study cannot be correctly interpreted without knowing why particular sequences reacted, additional characterization is needed. I would also consider putting at least some of these results in the main figures.

2) I am aware of several studies (such as Pitt and Ferré-D'Amaré, 2010 and Jiménez et al., 2013) in which high-throughput sequencing was used to characterize pools after in vitro selection experiments. In these studies, sequenced pools contained multiple copies of some variants, and a key component of the analysis was to show that copy number was correlated with catalytic or binding activity. I could imagine analyzing your experiments in a similar way. Furthermore, showing that some sequences appear multiple times in the high-throughput sequencing reads would provide additional evidence that they have enhanced ligations/recombination rates. I suspect that your evolved pools contained too many sequences to perform this type of analysis, but these issues should be at least briefly discussed.

3) It seems to me that a model of recombination in which a 5' hydroxyl group in one oligonucleotide attacks a phosphodiester bond in the other is simpler than the one proposed here (in which recombination requires both self-cleavage and ligation). I would recommend providing evidence from either experiments or the literature in support of your proposed mechanism.

4) RNA secondary structure can only be predicted to a limited extent. For this reason, I am not convinced that the ligation junctions of these sequences typically occur in the context of duplexes. Can you demonstrate this experimentally? Furthermore, I am not aware of reliable methods to predict RNA tertiary structure. For this reason (and because it is does not change any of the main conclusions of the paper) I would remove the statement in the Results section that J4 forms a purine-rich 4x4 internal loop with a triple-shared GA motif.

5) For me, one of the most exciting and interesting parts of the manuscript was your analysis of the reactivity of polymers other than RNA and DNA. However, the way in which the results were presented was not logical to me. I would recommend first comparing the ability of each of these polymers to promote ligation/recombination reactions under the same conditions, and then discuss new types of experiments (such as treating with base prior to incubation).

*Reviewer #2:*

Mutschler et al., have investigated spontaneous cleavage and ligation reactions among a diverse population of RNA (and RNA analogue) molecules. These reactions are well known, and the ability of a complementary template to accelerate the rate of ligation was first described over 40 years ago. Although these reactions have previously been carried out with mixed populations, this is the first study, to my knowledge, to apply deep sequencing to assess sequence preferences, especially for ligation. Not surprisingly, the authors find that there are some sequence preferences, which reflect a combination of intrinsic chemical reactivity and the potential for templated interactions. The reasons for the former are obscure, whereas the latter are due to both canonical and non-canonical templating effects. It is not surprising that DNA, ANA (arabino nucleic acid), and HNA (hexitol nucleic acid), all of which lack a vicinal cis-hydroxyl, do not undergo these reactions, whereas AtNA (altritol nucleic acid), which contains a vicinal cis-hydroxyl, does.

This is a well-executed study that will be of interest to nucleic acid chemists, but will not have broader appeal, as would be required for publication in *eLife*. The conclusions are generally well supported by the data. However, it is somewhat misleading to point to the "increase in the compositional and structural complexity of recombined pools" while ignoring the decrease in oligomer length that occurs due to the same reactions. If one considers only the ligated products, then those materials can be said to have increased complexity. But for the assembly of materials that involve both cleavage and ligation events (as in Figure 3), one must also consider the many shorter cleavage products that arise. It is not clear from this study how the reactions could achieve an overall increase in compositional complexity. Presumably this would require a fractionation process or means to shift the chemical equilibrium in favor of ligation. The present study would be more suitable for a specialized journal such as Nucleic Acids Research or Angewandte Chemie.

*Reviewer #3:*

This manuscript describes and extensively analyzes the ability of random sequences to ligate and recombine. The testing was primarily performed with RNA sequences and then compared across multiple genetic polymers (DNA, HNA, ANA, AtNA). The authors show (for the most part) that recombination via trans-ligation of two termini and via sequence displacement is detectable either directly using denaturing PAGE or, in cases when constant regions were added to the distal termini, using (RT)PCR. The reactions were performed in the eutectic phase of water-ice and control reactions incubated at -80 °C showed no activity. Overall this is a comprehensively designed and analyzed work with far-reaching importance, particularly for the Origin of Life field, but also for synthetic and chemical biology.

The ligated junctions for RNA seem to favor a G on the downstream of the ligation site. The RNA ligation site is thus biased, though not completely, towards a C/G sequence, not just CN. The authors use the unligated pool as a control; however, sequencing of unligated prey sequences is strongly biased by the method of introducing sequencing primers on the 5′ end (which depends on cross-templating by the RT enzyme and the SMARTER oligo substrate), so the statistics of the ligated vs unligated nucleotide at the 3′ side of the junction are not reliable. I understand the authors are being careful with their interpretation of the data, but the biases of the trans-templating method are known and can therefore be incorporated into the analysis, at least at the level of discussion if not directly in the Results section.

Figure 4B is key for the conclusions of this manuscript, but the bands for HNA are practically invisible and for AtNA the -80 control is very poorly visible. When intensified, both HNA and AtNA show bands corresponding to longer products. Ditto for DNA and ANA lanes in Figure Figure 4—figure supplement 1, where little to nothing is visible in either the experiment or the control lanes. These gels should be presented intensified as in Figure 3A. Related to this point, why are the -80 °C controls in Figure 4 invisible in the first place?

What are the diversities of libraries used? The theoretical limit of a 20-mer is 4^2^≈10^12^. Are these libraries exhaustive with respect to the potential diversity limit? Are there known biases in composition for the non-biological genetic polymers? The authors do not mention the amounts of material used, just concentrations. Clearly, sequencing of the entire starting pool (or even recombined and directly isolated ones) is not practical, but for the PCR-amplified pools, the sequencing results may be exhaustive. For that, however, an estimate of the starting diversity and the total amplification of the recombined products would have to be presented. The manuscript would be enhanced if these numbers were included for all experiments and pools, particularly for the RT-PCR-amplified pools, where the HTS results may represent nearly all ligated sequences.

The free energy of activation of a >p vs a triphosphate is known. Assuming similar activation energy for the ligation of a 5′-OH with >p and a triphosphate, the expected kinetics for the background reaction can be estimated from the Rohatgi and Szostak, 1996 paper. An estimate of the background ligation rate and acceleration by the individual motifs should be presented.

---

## [Author Response]

[Editors’ note: the author responses to the first round of peer review follow.]

[…] Reviewer #1:This study investigates the ability of random sequence pools of RNA, DNA, ANA, HNA, and AtNA to promote ligation and recombination reactions. The work is original, and explores important concepts related to the functional potential of polymers. However, from my perspective it could be significantly improved by considering the following issues.1) You used gel purification and high-throughput sequencing to identify molecules from different types of random sequence pools that undergo ligation or recombination reactions after long incubations. Each pool probably contains a small number of molecules with enhanced ligation or recombination rates and a large excess of sequences that react at background levels. After a single purification (analogous to a single round of selection), it is possible that these pools will still contain mostly inactive sequences. If this is the case, sequencing will not necessarily provide information about motifs that promote ligation/recombination (but see point 2). Figure 2—figure supplement 4 shows that you can measure the rates of some of these sequences in a reasonable amount of time, but not whether they are enhanced relative to arbitrary sequences or the starting pool. The fact that the sequence compositions of the ligated/recombined pools are different from the starting pools is encouraging in this respect. On the other hand, because the results of many experiments in this study cannot be correctly interpreted without knowing why particular sequences reacted, additional characterization is needed. I would also consider putting at least some of these results in the main figures.

We welcome this comment, which touches upon key aspects of our work and are addressing it below in detail.

The main criticism is based on a misinterpretation of the objective of the paper, which was not to bypass selection / SELEX strategies to isolate defined functional sequences (fitness peaks) (which has been done many times) but rather to investigate the global, systems-level fitness of random sequence pools as a whole (a measurement of which, to our knowledge has never before been attempted). In a sense, what we are trying to determine is what the reviewer 1 refers to as the ‘background’. In this respect, our main findings are the following:

1) Contrary to expectations, the global functionality of the pools exceeds what would be expected by extrapolation from SELEX-type experiments by orders of magnitude (see Figure 1, Figure 2, Figure 3, Figure 4 and Figure 5).

2) RNA pools in particular show an unprecedented innate and spontaneous propensity for recombination. Recombination increases the compositional, structural and informational complexity of RNA pools (as we demonstrate) and hence their adaptive potential (see Figure 4, Figure 5 and Figure 7).

3) Pools of other genetic polymers (specifically Altritol nucleic acids (AtNA)) not found in nature are also capable of recombination, establishing (1) that RNA is not unique in this regard and (2) revealing key chemical and structural parameters required for spontaneous recombination (see Figure 6).

We would argue that all three of these are significant findings with the potential to fundamentally change our perspective on early evolution and indeed on the functional potential of diverse oligomer pools in both biology and chemistry, and as such are of broad interest. However, we do accept that our arguments were not presented in sufficient breadth. In the revised manuscript, we have endeavoured to set out our findings both more clearly for the general reader and discuss them more fully within the context of previous work and the fundamental importance of the observed processes such as recombination in biology.

Please find below a more detailed examination of our main points:

I) in vitro selection experiments (SELEX etc.) have demonstrated repeatedly that diverse random sequence pools of RNA, DNA (as well as some unnatural XNA) oligomers comprise functional sequences including catalysts capable of RNA cleavage and ligation. We therefore may take it as a given that such functional sequences are present in those pools. However, various attempts at extrapolating the frequency of functional sequences all suggest that such functional sequences are extremely rare (10^-10^-10^-13^, see e.g. Bartel and Szostak, 1993; Wilson and Szostak, 1999 or Jiménez et al., 2013) depending on the study and functional criteria)). This is turn would suggest that such pools by themselves should be largely inert and that functional sequences would be impossible to discover in the absence of powerful selection methods involving i.e. efficient means for enrichment, replication and amplification. If true, this would clearly call into question one of the key conjectures about the origin of life, which posits a spontaneous emergence of function from random oligomer pools.

In the present work we set out to subject the above conjecture to a critical test based on the hypothesis that while individual functional sequences may be vanishingly rare, global pool functionality may be higher due to system-level interactions among pool sequences. Put simply, selection experiments identify individual sequences that are functional but discard simple interaction networks among pool sequences that may also be functional, e.g. as hetero-dimeric, -trimeric,. -oligomeric assemblies. Thereby, selection experiments likely underestimate the global phenotypic potential of the pools by an undue focus on individual sequences. Indeed, in some cases, as in our own selection experiments seeking to isolate improved RNA triplet polymerase ribozymes, the best adaptive solution turned out to be a heterodimer discovered fortuitously within our selection experiment designed to isolate the most active single sequences (Attwater et al., 2018).

II) Another argument in favour of considering the importance of intrapool interaction networks in potentiating global pool functional potential is their truly vast potential combinatorial diversity. For example, within the eicosamer pools examined herein, which encode a potential diversity of 10^12^ unique sequences, there are 10^12^ x 10^12^ = 10^24^ potential binary interactions (larger than Avogadro’s number!).

Indeed, the data described in our manuscript indicates that such pools are far from inert but display a pervasive tendency (of between 2-10% of pool sequences) for disproportionation reactions through transesterification chemistry resulting in ligation and recombination reactions among pool members.

III) As outlined above, our intent was never to “bypass” SELEX experiments but rather examine pools global functional potential. However, we accept that we may have given this impression inadvertently by our attempts to dissect the different RNA motifs contributing to pool functional potential in the case of RNA ligation via a 2’, 3’-cyclic phosphate intermediates. We now discuss these results more extensively and provide a more detailed breakdown of the different classes of reactive RNA molecules contributing to this result including an expanded Figure 2 and a new Figure 3 in the main text as suggested by the reviewer.

My major criticism of this work is that you did not show that these sequences react at enhanced rates relative to the average polymer sequence.

With regards to this specific point of reviewer 1, we would like to argue that we in fact did just that.

Our argument runs as follows:

1) The “average polymer sequence” is likely to be unreactive on the timescale examined (hence our proportions of 10% (Figure 1), respectively 2% (Figure 4) of reactivity (depending on pre-activation)). Hence, all ligated or recombined random or semi-random sequence pools show enhanced reaction rates compared to the “average” sequence.

2) Among these subsets of reactive sequences, we find that a majority of intermolecular ligation and recombination reactions proceed via a gapped helical intermediate (as had been described previously for templated >p ligation, see e.g. by Lutay et al., 2006, Biogeosciences (2006), 3: 243 or Usher et al., 1976, Science, 192: 53). In these, intermolecular hybridization provides an internal template and ligation as well as recombination transesterification chemistry via a >p intermediate is accelerated by proximity due to mutual sequence complementarity at the ligation junction.

To better illustrate point and the activity of these, we have now included data on four randomly picked sequence pairs as suggested by the reviewer. Of these, one pair (pair 1) is predicted to form a helical junction at the ligation site, whereas the other three sequence pairs (pairs 2-4) are predicted to form complexes but without a gapped duplex formation at the ligation junction. As expected, only pair 1 shows ligation under our experimental conditions (presumably ligation reaction promoted by proximity), while no ligation is observed for the other three pairs. This data is now shown in the new Figure 2—supplement 2.

3) This naturally leads to the question if all reactions would proceed via a gapped duplex intermediate or if among reactive sequence pairs we could identify other, distinct RNA motifs that would also accelerate intermolecular ligation. Indeed, bioinformatics analysis of a subset of reactive pool sequences allowed us to identify putative secondary structure elements promoting ligation, two examples of which (J4, H4) we analysed in more detail. As we show, J4 clearly accelerates cleavage / ligation beyond the rate provided by simple sequence complementarity and could therefore be referred to as nascent catalytic motif. H4 accelerates the reaction compared to the vast bulk of the pool sequences. However, while it is somewhat slower than a comparable helical intermediate, it promotes formation of a canonical 3’-5’ linkage, which is disfavoured in the helical arrangement (Lutay et al., 2006).

We would therefore argue that we have not only shown that some sequences react with an enhanced rate compared to the average sequence/background rate but that among these, there are again specific motifs, which react with even higher rates and/or distinct regiochemistry. More motifs are very likely to exist but their frequency in the subpool of sequenced RNAs was too low to allow their unambiguous identification.

In order to better illustrate the above arguments as suggested by the reviewer and outline this hierarchical analysis of RNA motifs involved in ligation, we have expanded this section of the manuscript including an expanded Figure 2 and a new Figure 3 in the main text.

The clearest way to address this point would be to experimentally measure the ligation/recombination rates of randomly chosen variants from at least one of the evolved pools. Are the ligation/recombination rates of these sequences higher than those of arbitrary sequences or the unevolved starting pool? Figure 2

As discussed above, there are potentially distinct levels of “background” reactivity. To better illustrate this point, we now specifically include analysis of two typical gapped duplexes as well as the two defined non-duplex RNA motifs identified by bioinformatic analysis. This data is now shown in the new Figure 2—supplement 2 and 6 and main text Figures 2 and 3.

With regards to reconstruction of oligomer reactivities, we note that we can only recapitulate bimolecular reactivity. Any sequences ligated by a 3rd partner oligonucleotide (e.g. acting as a splint) or via more complicated interaction networks or trajectories cannot be reconstructed as the participating oligonucleotides do not form part of the product. Thus, our analysis likely underestimates the extent and diversity of reaction trajectories and reactive motifs as our method is unable to sample in “trans-ligases”. Furthermore, our analysis also excludes non-canonical intra- or intermolecular ligation trajectories that yield circles, lariats or branched products. The lack of reactivity among some randomly picked sequence pairs in Figure 2—supplement 2 should be viewed in this context. Nevertheless, we show that salient features of reactivity as well as individual motifs can be extracted by careful analysis of the data.

Thus, our data unequivocally shows that random RNA sequence pools comprise a diverse range of functionalities both at the level of rare individual sequences (as was known before) and much more prominently at the systems-level / global pool level (which is a key result from the present work).

*2) I am aware of several studies (such as Pitt and Ferré-D'Amaré, 2010 and Jiménez et al., 2013) in which high-throughput sequencing was used to characterize pools after* in vitro *selection experiments. In these studies, sequenced pools contained multiple copies of some variants, and a key component of the analysis was to show that copy number was correlated with catalytic or binding activity. I could imagine analyzing your experiments in a similar way. Furthermore, showing that some sequences appear multiple times in the high-throughput sequencing reads would provide additional evidence that they have enhanced ligations/recombination rates. I suspect that your evolved pools contained too many sequences to perform this type of analysis, but these issues should be at least briefly discussed.*

We are of course aware of the work of Pitt and Ferré-D'Amaré, 2010 and Jiménez et al., 2013 but these studies differ from our work in that they sought to correlate the activity-sequence relationships in either a repertoire of mutants of a known small ribozyme motifs or in a population of selected anti-GTP aptamers. Thus, both of these studies sought to map fitness peaks centred around discrete sequences either identified previously or known to occur at high frequency upon selection in the pool examined.

In contrast, our work is aimed at a better understanding not of individual fitness peaks for a given activity but rather the global pool reactivity and potential for function (the fitness landscape in total) as it arises from systems-level interaction networks within the pools.

While we have also mapped reactivity to individual motifs, we did so as a means to better understand the underlying pool reactivity, not to map out fitness peaks.

With regard to sequencing coverage, reviewer 1 is indeed correct that, although there is ca. 1000-fold redundancy of 20-mer sequences in our 10^15^ sequence pools, the maximum available sequencing depth (10^9^) is insufficient to capture or map sequence enrichment without multiple rounds of selection. Taking this into account, we would therefore argue that the level of reactivity and identification of individual functional sequence pairs in our experiments is striking and unprecedented given the rarity of such sequences observed in previous selection experiments.

The latter is due to the fact that standard SELEX experiments are designed, to iteratively deplete and disrupt the interaction networks underlying global pool reactivity and instead isolate the exceedingly rate unitary sequences capable of function. While such experiments have been ground-breaking in revealing the potential of RNA (as well as DNA and various XNA) sequences for complex catalytic or ligand-binding functions, our results show that back-extrapolation of their frequency leads to a gross underestimation of pool functionality as only single sequences are considered.

Without wishing to overstretch the analogy, our results in many ways parallel similar findings in related fields such as chemistry, where increasingly systems-chemistry approaches are found to yield pathways to products believed to be impossible to obtain by conventional linear synthetic approaches.

In conclusion, in this manuscript we map for the first the time the innate, global functional potential of oligomer pools ab initio. Rather than seeking to isolate the “winners” in an evolutionary experiment (“race”) through cycles of selection, enrichment and amplification, we seek an overview of the potential of all “starters”, i.e. all pool sequences. We have now expanded the discussion of these points to set our work (and its motivation and conclusions) much clearer apart from “conventional” SELEX-type experiments.

3) It seems to me that a model of recombination in which a 5' hydroxyl group in one oligonucleotide attacks a phosphodiester bond in the other is simpler than the one proposed here (in which recombination requires both self-cleavage and ligation). I would recommend providing evidence from either experiments or the literature in support of your proposed mechanism.

We welcome this insightful comment. Previous studies examining RNA ligation via 2’, 3’-cyclic phosphate activation (>p) on an instructing template (e.g. Chetverin et al., 1997; Nechaev et al., 2009) suggested a 2-step mechanism (whereby a first step generates RNA oligomers terminating in >p by hydrolysis). Our own experiments (detailed in Figure 1) of RNA ligation in >p activated random eicosamer pools provides a baseline for reactivity via this chemistry in a random pool setting.

Our observation that recombination reactions in the same pools lacking >p proceeds on a ca. 10-fold slower timescale suggests that either a direct attack mechanism is kinetically slower or that recombination predominantly proceeds via a 2-step mechanism involving (1) (a slow step of) RNA hydrolysis generating >p and (2) ligation via >p. This has been our rationale for favouring this mechanistic trajectory over a direct in-line attack.

Nevertheless, reviewer 1 is of course correct that both mechanisms are likely to occur and indeed may be operating concurrently. In the light of this, we have amended the text to make this clear and now discuss both reaction trajectories in the context of ours as well as published findings.

4) RNA secondary structure can only be predicted to a limited extent. For this reason, I am not convinced that the ligation junctions of these sequences typically occur in the context of duplexes. Can you demonstrate this experimentally? Furthermore, I am not aware of reliable methods to predict RNA tertiary structure. For this reason (and because it is does not change any of the main conclusions of the paper) I would remove the statement on page 5 that J4 forms a purine-rich 4x4 internal loop with a triple-shared GA motif.

This is an important comment, but our arguments do not rely on accurate RNA tertiary (or even secondary) structure prediction.

We very much agree with the referee that exact topologies of RNA secondary structures (that ultimately determine the tertiary structure) are poorly predicted by most software packages (Zhao et al., (2018)). Physics-based free energy minimization approach of RNA secondary structure prediction has been evaluated in detail and found to have a base-pairing consistency of 70% between native and predicted secondary structures for moderately long RNAs (~70 - 500 nts, see e.g. Zhao et al., 2018; Doshi et al., 2004 or Mathews et al., 1999).

However, for our raw analyses only the accuracy of the base-pairing matters (where 70% accuracy is sufficient for a global picture) while “average” tertiary structures are not (and cannot) be analysed using this method. Furthermore, our finding that >p ligation is favoured at helical junctions is not only in full agreement with the literature (Lutay et al., 2006; Usher et al.,1976; Bolli et al. 1997), but we have provide experimental verification in individual examples in this manuscript (Figure 2—figure supplement 2, Figure 2—figure supplement 4, Figure 2—figure supplement 6) suggesting that at the level of the comparatively short RNA oligomers examined herein, predictions are largely trustworthy.

Finally, we note that our suggestion that the J4 motif folds into a 4x4-loop with a triple-sheared GA motif derives not from bioinformatic predictions, but from the fact that J4-like sequence motifs are found in several natural RNAs, for some of which high-resolution structural information is available (23S rRNA of *Deinococcus radiodurans* (PDB ID: 2ZJR) / *Escherichia coli* adenosylcobalamin riboswitch (PDB ID: 4GMA)). These structures provide a rationale as to why and how J4 may promote ligation via a 2’, 5’ linkage (as shown in the new Figure 3). Thus, our statement is backed up by multiple lines of structural evidence and we have therefore not altered it in the revised manuscript. Nevertheless, we now provide more context and discussion on the structural folds of J4 and the other RNA motifs to present our arguments more clearly including main text Figure 2, Figure 3 and Figure 2—figure supplement 3, Figure 2—figure supplement, Figure 2—figure supplement 3, Figure 2—figure supplement 4, Figure 2—figure supplement 5 and – Figure 2—figure supplement 6.

5) For me, one of the most exciting and interesting parts of the manuscript was your analysis of the reactivity of polymers other than RNA and DNA. However, the way in which the results were presented was not logical to me. I would recommend first comparing the ability of each of these polymers to promote ligation/recombination reactions under the same conditions, and then discuss new types of experiments (such as treating with base prior to incubation).

We welcome this comment and agree that the differential reactivity of chemically closely related but distinct genetic polymers are a key finding.

As suggested by reviewer 1, we had originally compared all genetic polymers under the same conditions (e.g. neutral pH, 10 mM Mg^2+^), where we only observed reactivity with RNA pools (Figure 6A). We subsequently introduced modified conditions in order to expedite reactivity in the inert pools e.g. addition of Zn^2+^ in the case of DNA and ANA pools, which had previously been shown to be an essential cofactor in autocatalytic cleavage by small DNA motifs (Gu et al., 2013). However, we found that even in the presence of Zn^2+^, we observed no reactivity in the DNA (or ANA) pools. These negative results are now provided in Figure 6—figure supplement 1.

As we had already observed degradation of AtNA (but not HNA) in response to alkaline pH (although 6-fold slower rate for AtNA than RNA, Figure 6—figure supplement 2), we tested RNA, HNA and AtNA pool reactivity following a pulse of NaOH (now Figure 6B). Indeed, this revealed that AtNA, like RNA- is able to recombine, although reactivity was noticeably weaker (ca. 10-20-fold). In contrast, we did not observe recombination of AtNA (and HNA) in absence of NaOH. We had previously not included these negative results but now the data is presented in the new Figure 6—figure supplement 5. A direct comparison of the observed recombination yields between RNA and AtNA is also complicated by the fact that observed yields depend on the efficacy of the respective reverse transcriptase, the sensitivity of which undoubtedly better for RNA. We have now expanded the discussion of the above results in the revised manuscript to better outline the rationale for our different experimental strategies.

Reviewer #2:Mutschler et al., have investigated spontaneous cleavage and ligation reactions among a diverse population of RNA (and RNA analogue) molecules. These reactions are well known, and the ability of a complementary template to accelerate the rate of ligation was first described over 40 years ago. Although these reactions have previously been carried out with mixed populations, this is the first study, to my knowledge, to apply deep sequencing to assess sequence preferences, especially for ligation. Not surprisingly, the authors find that there are some sequence preferences, which reflect a combination of intrinsic chemical reactivity and the potential for templated interactions. The reasons for the former are obscure, whereas the latter are due to both canonical and non-canonical templating effects. It is not surprising that DNA, ANA (arabino nucleic acid), and HNA (hexitol nucleic acid), all of which lack a vicinal cis-hydroxyl, do not undergo these reactions, whereas AtNA (altritol nucleic acid), which contains a vicinal cis-hydroxyl, does.1) This is a well-executed study that will be of interest to nucleic acid chemists, but will not have broader appeal, as would be required for publication in eLife.

We would like to respectfully disagree with this assertion of reviewer 2. Recombination, the exchange of information between different genetic polymer strands, is of fundamental importance in biology (see e.g. Visser and Elena, 2007 Pesce et al., 2016) both for genetic diversification in the germline (meiosis) and soma (e.g. gene conversion in avian antibody generation) as well as genome maintenance and repair. It is furthermore of critical importance to guard against the progressive corruption of genetic information by the deleterious effects of genetic drift (i.e. Muller’s ratchet), enabling elimination of neutral or weakly deleterious mutations by backcrossing (Muller, (1964)).

RNA recombination is widespread among RNA viruses (Simon-Loriere and Holmes, 2011). While generally attributed to template switching etc. recombination by inferred purely chemical processes has been observed in various viruses such as Poliovirus (Gmyl et al., 1999), rubella virus (Adams et al., 2003) or Qbeta bacteriophage (Chetverin et al., 1997), pointing to the possibility of a wider importance of non-enzymatic RNA recombination in present day biology with implications for non-viral transcriptome complexity and fluidity.

We would therefore argue that our finding that RNA – due to its specific chemical constitution- has an innate capacity and tendency to recombine even in the absence of recombinase enzyme or extraneous activating chemistry has important implications both for origins research as well as indeed the analysis of all diverse RNA oligomer assemblies.

In the context of primordial RNA pools, this innate capacity has fundamental implications as it provides for a progressive disproportionation of RNA sequences, which does increase pool complexity by the de novo generation of diversity of RNA oligomer length and sequence content and secondary structure driven by intrapool RNA base pairing networks (see also response to point 2 below).

The potential for spontaneous recombination is likely innate to any diverse assembly of RNA sequences (including those found in biology) and should be considered when analysing such assemblies and their dynamics including studies of the transcriptome and viral populations and quasi-species.

2) The conclusions are generally well supported by the data. However, it is somewhat misleading to point to the "increase in the compositional and structural complexity of recombined pools" while ignoring the decrease in oligomer length that occurs due to the same reactions. If one considers only the ligated products, then those materials can be said to have increased complexity. But for the assembly of materials that involve both cleavage and ligation events (as in Figure 3), one must also consider the many shorter cleavage products that arise. It is not clear from this study how the reactions could achieve an overall increase in compositional complexity.

We welcome this insightful comment by reviewer 2 and agree that our claims of increased compositional and informational diversity/complexity need to be supported by a more comprehensive analysis of the impact of recombination on pool complexity.

1) We would like to point out that while notions of spontaneous generation of increased informational and compositional complexity by RNA pools may seem counterintuitive, they are in fact both predicted and supported by theoretical arguments from statistical mechanics and thermodynamics, which suggest that under the right boundary conditions an increase in length and informational complexity of (RNA) polymer chains is favoured by entropic factors (see e.g. Blokhuis and Lacoste, 2017). We now outline these arguments as part of an expanded Discussion section in the revised manuscript.

2) We now provide in silico simulations and analysis of the population level Shannon entropy (an established measure of pool sequence diversity i.e. informational complexity (Derr et al., 2012)) of pre- and post-recombination pools. These calculations consider total pool informational content, under conditions of random oligomer cleavage and ligation and show that such recombination indeed increases the pool informational diversity under almost any boundary condition, provided that recombination is applied to a pool with significant sequence redundancy. Indeed, this applies to our experimental system: the 10^15^- member eicosamer pools analysed herein are redundant with respect to encoded information, covering the total combinatorial diversity (4^20^ =1.09 x 10^12^) ca. 1000-fold (i.e. there are approx. 900 copies of each eicosamer sequence within the pool).

Our finding that recombination diversifying parts of a redundant sequence pool generates new information is also intuitively understandable, as recombination of part of a redundant pool generates new sequences (both in length distribution and information content), while maintaining (i.e. not destroying) the previous sequence and information content (as redundancy persists but is reduced). In other words, sequence / compositional / informational complexity is increased at the expense of redundancy.

Other outcomes of our calculations may be less obvious but no less interesting. These include the finding that the informational gains through recombination scale both with sequence code complexity (4 letter code > 2 letter code) and oligomer length as Recombination of longer oligomer pools generates more diversity. The simulations illustrate the trade-offs between enhanced cleavage rates, which generate more short fragments (loss of information) but also generate more ligatable fragments (and thus enable information gains) and ligation rates, which fix informational gains.

In addition to detailed calculations, we now also provide an extended discussion of these points in our revised manuscript including a new Figure 7 and accompanying figure supplements.

3) It is not clear from this study how the reactions could achieve an overall increase in compositional complexity. Presumably this would require a fractionation process or means to shift the chemical equilibrium in favor of ligation. The present study would be more suitable for a specialized journal such as Nucleic Acids Research or Angewandte Chemie.

Reviewer 2 points out an important aspect of our study. While the above (see point 2) simply examines the generation of diversity as a theoretical concept of informational complexity in the context of early evolution, the impact of recombination on sequence pools under fractionation and / or adaptive pressure is more significant and more relevant to real world scenarios.

Indeed, there are a number of physicochemical processes (including thermophoretic cycles in porous substrates / adsorption to (mineral) surfaces etc.), which would favour length-extended oligomers over pool members or short cleavage products (see for example Kreysing et al., 2015). However, the simplest fractionation process that favours more diverse and length extended oligomer sequences is selection for (higher order) function.

Previous studies had established the importance of pool diversity as well as increased oligomer length not only for the discovery of functional motifs but, more importantly, for the discovery of the most active or complex functional motifs (reviewed in Pobanza and Luptáka, 2016), even in the presence of trade-off from an increased tendency for ambiguous folding of longer RNA oligomers (Sabeti et al., 1997). Furthermore, the power of continued recombination of sequence pools under selective pressure to accelerate adaptive walks and evolutionary optimization is clearly demonstrated both by examples from biology (gene conversion, RNA virus recombination (see above)) and protein engineering (see e.g. Stemmer, 1994, reviewed in Arnold, 2009).

Therefore, we would argue that this all leads to an inescapable conclusion, which is that the innate potential of RNA oligomer pools for recombination provides a route unique to RNA (and to some degree other genetic polymers, see below) to bootstrap themselves towards higher informational, compositional and structural diversity and hence evolutionary potential.

Given the challenges of prebiotic chemistry to generate oligomer pools of substantial length (the here investigated 20-mers are very much at the upper end of RNA oligomer lengths reported containing all four nucleotides) the innate potential of such pools to extend themselves in length and diversity by recombination is highly significant. In light of theoretical arguments, which suggest a sharp drop in the potential for folding into stable secondary (and by inference tertiary) structures needed for function below a certain oligomer length limit (Briones et al., 2009), recombination might have been of fundamental importance for the emergence of the first functional RNAs.

Furthermore, in the presence of a fractionation process selecting for increased oligomer length (e.g. thermophoresis (see above)) or a given function (such as binding to a substrate and / or catalysis), RNA pools capable of recombination would yield a continuous supply of extended, more structured and more functional motifs and thereby outcompeting putative rival genetic polymers (such as DNA) unable to recombine. Such process would be potentiated by an ongoing process of recombination during selection (or cyclical iteration of either) as recombination has previously been shown to be a powerful means to accelerate adaptive walks (see above).

Finally, having established the potentially profound importance of recombination for the emergence of function from oligomer pools, our work begins to define key chemical parameters required for the display of an innate potential for recombination. Among these, a vicinal diol geometry on the ring appears to be crucial enabling formation of a cyclic phosphate upon hydrolysis, which supports subsequent ligation.

Reviewer #3:[…] The ligated junctions for RNA seem to favor a G on the downstream of the ligation site. The RNA ligation site is thus biased, though not completely, towards a C/G sequence, not just CN. The authors use the unligated pool as a control; however, sequencing of unligated prey sequences is strongly biased by the method of introducing sequencing primers on the 5′ end (which depends on cross-templating by the RT enzyme and the SMARTER oligo substrate), so the statistics of the ligated vs unligated nucleotide at the 3′ side of the junction are not reliable. I understand the authors are being careful with their interpretation of the data, but the biases of the trans-templating method are known and can therefore be incorporated into the analysis, at least at the level of discussion if not directly in the Results section.

We welcome this perceptive comment.

The bias of trans-templating is of course real and similar issues exist in other library preparation methods for diverse nucleic acid pools. However, we have several lines of evidence that the CpN dinucleotide signature is a direct result of the transesterification reaction. These are detailed below:

1) We prepared ligated and recombined RNA pool fractions using a range of different methods (incuding adaptor ligation, direct RT-PCR, SMARTER RT-PCR) but observed the CpN bias regardless. Even the N20 x N20A10 products produced by polyT-primed SMARTer RT-PCR (Figure 5), which we do not compare to preligation material show a raw C-bias at the NpN recombination site in products ≤ 50 nts (Figure 5C). Here, the random nucleotide at the position of the recombination event was never exposed to terminal adapter ligation or template switching.

2) The reference pools of unligated N_20_>p and C1/C3-bait>p semi-random RNA (controls for experiments described in Figure 1 respectively Figure 2) were prepared by dephosphorylation of the >p followed by adapter ligation of a preadenylated oligonucleotide and not by SMARTer RT-PCR. While this protocol is also prone to sequence biases at the 3’-end (and close to it), due to sequence preferences of the ligase, the reference pools used in Figure 1-supplement 1 are unbiased further upstream of the adapter-ligated 3’-end, where the comparison happens and where we observe the same CpN signature in all recombination reactions. Furthermore, even without taking the pre-ligation material as reference, (and as already stated in 1), a raw C-bias is directly visible in all recombination products shown in the colour plot in Figure 4D, Figure 5C and the supplementary movie, where the C-bias follows the recombination “seam” of the differently sized products independently from the library preparation method.

Thus, the overall ligation junction sequence bias / motif CpN was apparent throughout all RNA recombination experiments with some (minor) variations as detailed in Figure 1, Figure 4, Figure 5 and Figure 1—figure supplement 1. While referee 3 is correct at noting the bias for a downstream G (CpG) in the N_20_>p experiments, this bias is not observed in all cases (e.g. the semi-random RNAs shown in Figure 1—figure supplement 1 and Figure 2—figure supplement 1D) and we have therefore not elaborated on it.

Figure 4B is key for the conclusions of this manuscript, but the bands for HNA are practically invisible and for AtNA the -80 control is very poorly visible. When intensified, both HNA and AtNA show bands corresponding to longer products. Ditto for DNA and ANA lanes in Figure 4—figure supplement 1, where little to nothing is visible in either the experiment or the control lanes. These gels should be presented intensified as in Figure 3A. Related to this point, why are the -80 °C controls in Figure 4 invisible in the first place?

The reviewer is correct that bands for HNA (both -9 ^o^C & -80 ^o^C), AtNA (-80 °C), DNA (both -9 ^o^C & -80 ^o^C) and ANA (both -9 °C & -80 °C) are practically invisible. Indeed, these reactions are what we consider negative (i.e. ligation / recombination did not take place, and hence no unambiguous PCR amplicons (larger than primer-dimers) were detected (in contrast with clear amplification products of RNA (-9 ^o^C) and AtNA (-9 ^o^C)).

As reviewer 3 correctly observes, it is also possible to make out faint bands in some of the other reactions including negative control PCRs, e.g. at -80 °C. However, none of these weak bands relate to (weak) recombination / ligation activity in e.g. the DNA, ANA or HNA pools for a number of reasons:

The appearance of weak amplification products is a common occurrence in all PCR reactions, where (spurious) bands will eventually appear after sufficient PCR cycles, as even minute amounts of mispriming or cross-contamination are exponentially amplified. The problem is greatly increased in PCR amplifications from random (or semi-random) sequence pools due to the increased likelihood of mispriming events on the random sequence segments. Furthermore, the weak bands pointed out by the referee occur to the same extent in both the -9 °C samples (incubated for prolonged periods) and -80 °C negative controls (which were immediately deep frozen at -80^o^C) and can therefore not be the consequence of a reaction occurring during the incubation period.

The typical low-molecular weight band seen in the -80 °C negative control reactions (likely due to mispriming of the RT-primer on the random sequence part of the bait fragment) is reduced or absent in Figure 6B (previously Figure 4B), due a) to the pre-incubation NaOH treatment, which leads to fragmentation of semi-randomers (in the case of RNA and AtNA) and b) a lower sensitivity of the RT enzyme for HNA and AtNA. Both a) and b) cause less opportunity for spurious background mispriming of the RT primer in the experiment shown in Figure 6B. At the same time, the intensity of the *bona fide* recombination products in both RNA (and AtNA) (when incubated at -9 ^o^C) is enhanced by NaOH (as would be expected).

We therefore stand by our previous conclusion that only RNA (and to a lesser extent AtNA) pools have an innate potential for ligation and recombination while this capacity is either absent or negligible in DNA, ANA and HNA pools.

What are the diversities of libraries used? The theoretical limit of a 20-mer is 4^2^≈10^12^. Are these libraries exhaustive with respect to the potential diversity limit? Are there known biases in composition for the non-biological genetic polymers? The authors do not mention the amounts of material used, just concentrations. Clearly, sequencing of the entire starting pool (or even recombined and directly isolated ones) is not practical, but for the PCR-amplified pools, the sequencing results may be exhaustive. For that, however, an estimate of the starting diversity and the total amplification of the recombined products would have to be presented. The manuscript would be enhanced if these numbers were included for all experiments and pools, particularly for the RT-PCR-amplified pools, where the HTS results may represent nearly all ligated sequences.

We thank reviewer 3 for this insightful comment and apologize for having omitted a fuller discussion of the diversities explored, which we now provide in the revised manuscript.

We chose eicosamer pools because, within the repertoire sizes explored within our experiments (10^14^-10^15^ molecules), there is considerable redundancy within these pools (diversity 4^20^ = 1.09 x 10^12^). Although we are unable to completely capture this within the current limits of NGS sequencing (ca. 10^9^ reads), we can infer whole pool diversity from the degree of randomness of the partially sequenced pool.

Even within reacted pools where we observe 10% (Figure 1, 10^14^ sequences) resp. 2% or 0.2% (Figure 5, 10^13^ or 10^12^) of pool sequences ligated / recombined, NGS cannot provide full coverage. We are therefore, in all cases, observing a representative sample. Within that sample we can exploit similarities and redundancy such as is present in structural classes of RNA motifs. We leverage this in the bioinformatic RNA motif classification analysis we present (Figure 2, Figure 3 and associated supplementary figures) and show that it enables us to identify reactive motifs.

Another issue worthy of consideration is the fact that while we obtain RNA and DNA pools directly by solid-phase synthesis), XNA pools (ANA, HNA, AtNA) need to be obtained by enzymatic synthesis using bespoke engineered polymerases, as solid phase synthesis is not available. While enzymatic synthesis is efficient on random DNA templates, inevitable biases are introduced, which reduce the diversity of XNA pools compared to DNA and RNA pools. Nevertheless, this is unlikely to significantly impact pool reactivity, given the considerable redundancy within eicosamer pools.

The free energy of activation of a >p vs a triphosphate is known. Assuming similar activation energy for the ligation of a 5′-OH with >p and a triphosphate, the expected kinetics for the background reaction can be estimated from the Rohatgi and Szostak, 1996 paper. An estimate of the background ligation rate and acceleration by the individual motifs should be presented.

We welcome this perceptive comment. However, we would like to refrain from using triphosphate dependent ligation as a reference as the activation energy is critically dependent on the presence and concentration of divalent metal ions like Mg^2+^ (which are absent in the majority of our reactions). Furthermore, in triphosphate-based ligations, the diphosphate leaving group renders the ligation reaction quasi-irreversible (k_obs_ = k_ligation_). In contrast, >p-dependent ligation is reversible due to the absence of a leaving group (k_obs_ = k_ligation_ + k_cleavage_) and the (chemically) almost isoenergetic reaction equilibrium.

Finally, it is challenging to extrapolate triphosphate-dependent ligation rates determined under ambient in-solution conditions to reactions taking place at much lower temperatures in eutectic in-ice conditions.

In order to provide a (better) estimate of the rate acceleration, we now compare the apparent reaction rates (k_obs_) for motif J4 and H4 to the k_obs_ of three template / splinted ligation reactions (which may be considered the background ligation rate occurring in a gapped duplex) and summarize this data in the new Figure 2—figure supplement 6.